

# Uncertainties in the spatial distribution of snow sublimation in the semi-arid Andes of Chile

Marion Réveillet[1], Shelley MacDonell[1], Simon Gascoin[2], Christophe Kinnard[3], Stef Lhermitte[4] and Nicole Schaffer[1]

[1]Centro de Estudios Avanzados en Zonas Áridas (CEAZA), ULS-Campus Andrés Bello, Raúl Britan 1305, La Serena, Chile.

[2]Centre d'Etudes Spatiales de la Biosphère (CESBIO), Université de Toulouse, CNRS/CNES/IRD/INRA/UPS, 31400 Toulouse, France.

[3]Département des Sciences de l'Environnement, Université du Québec à Trois-Rivières, 3351 Boul. des Forges, Trois-Rivières, QC, G9A5H7, Canada.

[4]Department of Geoscience & Remote Sensing, Delft University of Technology, Delft, The Netherlands

*Correspondence to*: Marion Réveillet (marion.reveillet@ceaza.cl.)

**Abstract.** In the semi-arid Andes of Chile, farmers and industry in the cordillera lowlands depend on water from snowmelt, as annual rainfall is insufficient to meet their needs. Despite the importance of snow cover for water resources in this region, understanding of snow depth distribution and snow mass balance is limited. Whilst the effect of wind on snow cover pattern distribution has been assessed, the relative importance of melt versus sublimation has only been studied at the point-scale over one catchment. Analyzing relative ablation rates and evaluating uncertainties are critical for understanding snow depth sensitivity to variations in climate and simulating the evolution of the snow pack over a larger area and over time. Using a distributed snowpack model (SnowModel), this study aims to simulate melt and sublimation rates over the instrumented watershed of La Laguna (3150–5630 m above sea level, 30°S), during two hydrologically contrasted years. The model is calibrated and forced with meteorological data from nine Automatic Weather Stations (AWS) located in the watershed, and atmospheric simulation outputs from the Weather Research and Forecasting (WRF) model. Results of simulations indicate first a large uncertainty in sublimation ratios depending on the forcing. The melt/sublimation ratios increased by 100% if forced with WRF compared to AWS data due to the cold bias and precipitation over-estimation observed in WRF output in this region. Second, the simulations indicate similar sublimation ratios for both years, but ratios vary with elevation with a relative increase in melt at higher elevations. Finally results indicate that snow persistence has a significant impact on the sublimation ratio due to



higher melt rates.

# 1. INTRODUCTION

In the semi-arid Andes, the cryosphere is the dominant water source, as rainfall is episodic and insufficient to meet user demand. The region is characterized by very low precipitation amounts that are largely limited to winter months, and are intermittent. Large interannual variability is observed, as the area is strongly affected by El Nino South Oscillations (ENSO) (e.g. Falvey and Garreaud, 2007; Garreaud, 2009; Montecinos et al., 2000). In broad terms, during El Niño periods the semi-arid Andes are characterized by warm air temperatures and higher precipitation totals, whereas La Niña periods are on average colder with less precipitation (e.g. Ducan *et al.,* 2008). Whilst snowmelt comprises the bulk of available water (Favier et al., 2009), due to low humidity, high solar radiation and strong winds, sublimation is a significant ablation process, especially at high elevations (Ginot et al., 2001; Gascoin et al., 2013; S. MacDonell et al., 2013). Consequently, quantifying snow mass balance processes are crucial for predicting current water supply rates, and for informing future projections.

Despite the importance of snow cover for water resources in this region, there is currently a limited understanding of snow depth distribution and mass balance, largely due to the difficulty of accurately measuring and modeling both accumulation and ablation processes in this area (Gascoin et al., 2011). Temperature index models have been shown to be inadequate to evaluate mass balance processes in the semi-arid Andes, due to the importance of the latent energy flux (Ayala et al., 2017). However, an energy balance model requires a larger input dataset that is often not available in Andean catchments due to the logistical difficulty of Automatic Weather Station (AWS) installation and maintenance. Therefore, the evaluation of alternative methods for acquiring distributed meteorological information is required. Options include the use of interpolation/extrapolation strategies (e.g. MicroMet, Liston and Elder, 2006b), reanalysis (NCEP, Kalnay et al., 1996) or atmospheric model outputs (e.g. Weather Research and Forecasting (WRF) model, Skamarock and Klemp, 2008). For the semi-arid Andes, both MicroMet extrapolation based on AWS data (Gascoin et al., 2013) and atmospheric models (e.g. Favier et al., 2009; Mernild et al., 2017) have been used to force snow model evolution. However, none of these studies have quantified the uncertainties related to forcing data.

 The relative importance of melt and sublimation to total ablation has been studied at both the point-scale (MacDonell *et al.,* 2013) and catchment scale (Gascoin *et al.,* 2013) in one catchment in the semi-arid Andes. MacDonell *et al.* (2013) estimated that the sublimation fraction was 90% at high altitude (>5000m a.s.l.) in an extreme environment with predominantly sub-freezing temperatures and strong local wind speeds. Using a distributed snowpack model, Gascoin *et al.* (2013) found that the total contribution of sublimation (static-surface and blowing snow sublimation) to total ablation in the Pascua-Lama area (29.3° S, 70.1°W; 2600 - 5630 m a.s.l.)





was 71%. However, this value was obtained for one snow season, and the precipitation was estimated from snow depth measurements as precipitation gauge data were unreliable. The sensitivity of sublimation to meteorological forcing and in particular to precipitation uncertainties was not evaluated.

The objective of this study is to assess the uncertainties related to modeling snow evolution in the semi-arid Andes using AWS and WRF-model generated meteorological datasets during two contrasting years. From this analysis, the snow mass balance for relatively wet and dry years will be compared, and an evaluation of the impacts of model choices on sublimation and melt rates in dry mountain areas will be discussed.

To address this aim, the model SnowModel described in Liston *et al.* (2006) will be applied to the La Laguna catchment in the semi-arid Chilean Andes during 2014 and 2015. These two years were selected because in this region, 2015 was considered to be a strong *El Nino* event, associated with warm and wet conditions, whereas 2014 was drier and colder and considered a neutral year (Ceazamet; http://origin.cpc.ncep.noaa.gov). We hypothesize a larger sublimation ratio for winter 2014 than 2015, due to drier and cooler conditions which would inhibit melt. Conversely, higher precipitation totals in 2015 would lead to increased snow depths and snow persistence over time, favoring melt and therefore decreasing the sublimation ratio.

## 2. STUDY SITE AND DATA

### 2.1 Study site

La Laguna watershed is located in the semi-arid Andes of Chile in the Elqui Valley (30°S), 200km East of La Serena, close to the border with Argentina (Figure 1a). As it is easily accessible this catchment is the best instrumented with an unusually high density of AWS, especially during 2014 and 2015.

The catchment covers an area of ~513 km$^2$ and elevations range from 3150 to 5630 m a.s.l. (Figure 1b). At these elevations, only minimal vegetation in the form of shrubs is observed, so we do not consider vegetation in this study. The study area includes rock glaciers and clean glaciers. Tapado Glacier is the largest of these with an area of 2.2 km$^2$ (Figure 1b). This catchment was selected since it is an important water resource in the Elqui valley. Indeed it feeds water to the La Laguna reservoir (200 Mm$^3$ capacity), which is part of the strategic irrigation system in the Elqui Valley. Nevertheless the precipitation amount is very low. The mean annual rate measured at la Laguna station is ~200 mm a$^{-1}$ and precipitations events are episodic with less than 10 events per year. Most of these events (90%) occur during the winter period (Rabatel et al., 2011), as snow fall. This seasonal difference is mainly due to differences in the position and intensity of a high-pressure cell in the eastern Pacific Ocean. During the summertime the high-pressure cell limits advection, while during the winter it moves further North, allowing the moisture-laden depressions to reach the area (Garreaud et al., 2011). The temporal variability of precipitation



is also complicated by precipitation trajectories which impact the frequency of precipitation events (e.g. Sinclair and MacDonell, 2016).

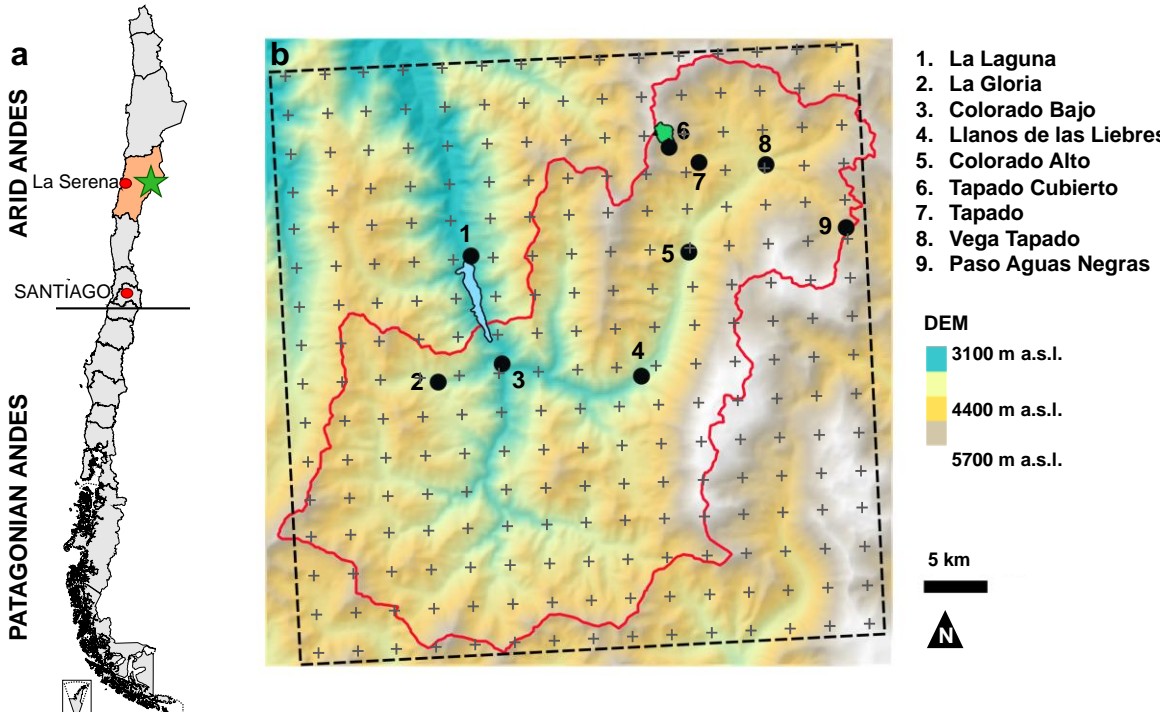

*Figure 1: a) Map of Chile with Coquimbo Region colored orange and the catchment location identified with a star. b) DEM (SRTM, 100m) of La Laguna catchment. Red line corresponds to the catchment delineation, black dashed line to the WRF domain containing the virtual WRF stations (grey crosses). Blue area is La Laguna reservoir and greed area is the Tapado Glacier. AWS stations are the 9 black points.*

## 2.2. Data

### 2.2.1 Digital elevation model

The digital elevation model (DEM) used in this study was derived from the CGIAR hole-filled 3 arc-second SRTM DEM resampled to 100 m resolution by the cubic method. The 100 m resolution was chosen to facilitate alignment of the model grid with the 500 m resolution MODIS products (see below).





### 2.2.1 Meteorological data

**a/ Automatic weather station measurements**

Meteorological data from nine Automatic Weather Stations (AWS) are available in the catchment (Figure 1b) over the study period 2014-2015. La Laguna, Tapado and Paso del Agua Negra AWSs are permanent stations

maintained by CEAZAmet (www.ceazamet.cl) with hourly measurements. In addition, five HOBO® weather stations (Colorado Bajo, La Gloria, Llano de Las Liebres, Colorado Alto and Vega Tapado) were installed in March 2014 and set to record meteorological data every 30 minutes. Finally, the Tapado Cubierto station was installed in 2013 for a specific campaign and provides measurements at hourly intervals. More details regarding available measurements and the time periods for which these are available are provided in Figure 2. Tapado

records are reported in Figure 3 as an example of the weather conditions in this catchment.

Due to the complexity of precipitation measurement (e.g. MacDonald and Pomeroy, 2007), datasets were post-processed. First, filters were applied to eliminate outliers (i.e negative values and values larger than 30 mm.h$^{-1}$). Second, satellite images (MODIS Aqua and MODIS Tierra) were used to remove recorded precipitation events on sunny days, which were probably due to wind transport. These precipitation events were only removed if five

cloud-free images were available (i.e. the two for the day: one the afternoon before and one the next morning). In total, three precipitation events lower than 2 mm w.e. were removed with this method.

At Tapado, measurements are recorded at two Geonor weighing precipitation gauges, of which one is shielded (Alter Shield) and one is unshielded. After being filtered, the cumulative difference at the two gauges was 9.1 mm for 2014 (i.e. ~10%; 97.1 mm for unshielded gauge 1 and 106.2 mm for shielded gauge 2) and 5.4 mm for

2015 (i.e. ~1% ; 457.5 mm for gauge 1 and 462.9 mm for gauge 2) with a maximum hourly difference of 1.1 mm, however the relative bias between the sensors is neither constant nor unidirectional. As the difference between the two sensors was relatively small, the mean of the two dataset was used as the reference precipitation value, and a maximum uncertainty of 10% was estimated.



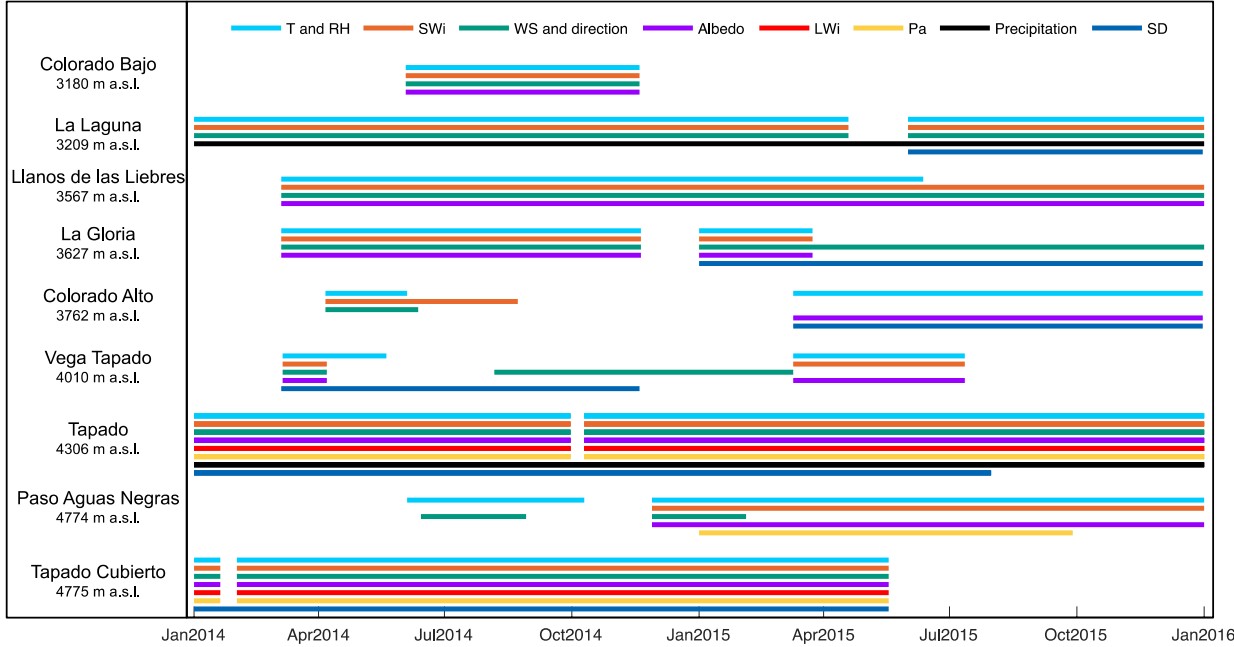

*Figure 2: Date of available measurements from the nine AWSs used to calibrate and validate the model. T is the air temperature, RH the relative humidity, SWi and LWi the incoming short and long wave radiation respectively, WS the wind speed and Pa the atmospheric pressure and SD the snow depth.*







*Figure 3: Example of climate conditions of the study area. Hourly air temperature (T), relative humidity (RH), incoming shortwave (SW) and longwave (LW) radiation, air pressure (Pa) precipitations (P) and wind speed and direction (wind roses) measured at the Tapado AWS from the $1^{st}$ of January 2014 to the $31^{st}$ of December 2015.*



## b/ WRF model outputs

The Weather Research and Forecasting (WRF) model was used to force SnowModel. WRF is usually used to predict weather for atmospheric research and operational weather forecasting. Using reanalysis-data as boundary conditions, this model is able to provide hourly meteorological data such as air temperature, relative humidity, incoming radiation, wind speed and direction, atmospheric pressure and precipitations. In this study WRF was forced by 6 hourly data from the National Center for Environmental Prediction (NCEP) reanalysis data, at 1° grid resolution (Kalnay et al., 1996), as well as daily surface temperature of the ocean at 0.083° of resolution. WRF model has been run using the model version 3.7.1 (Skamarock and Klemp, 2008), with default parameterizations (this choice is discussed in section 5.1). The model has been run over a time period covering the entire study period (from April 2013 to April 2016). The model outputs are available at ~22 km resolution over the entire Chile, ~7 km resolution at the region scale, and 3 km resolution over La Laguna the catchment. The hourly 3 km outputs (T, RH, WS and direction, P, SW, LW, Pa) over the catchment (Figure 1) represent the virtual stations that have been used in this study.

## 2.2.3 Local snow depth data

In 2014, three AWSs provided snow depth measurements at an hourly time step (Vega Tapado, Tapado and Tapado Cubierto; Figure 2). Over the 2015 winter, snow depth measurements were available at five stations (La Gloria, Colorado Alto, Tapado, La Laguna and Tapado Cubierto). The snow depths were measured with ultrasonic sensors and require post-treatment because they are particularly prone to measurement errors and typically produce a noisy signal (Lehning et al., 2002). Therefore, a control procedure described in Lehning et al., (2002) has been applied to clean the signal, and in particular to eliminate spikes, check for outliers and physical limits.

## 2.2.4 MODIS snow products

MOD10A2 (Terra) and MYD10A2 (Aqua) snow products version 5 were downloaded from the National Snow and Ice Data Center (Hall et al., 2006; Hall and Riggs, 2007) for the period 1 January 2014 – 1 January 2016. The binary snow products were projected on a 500 m resolution grid in the same coordinate system as the DEM. Missing values, mainly due to cloud obstruction, were interpolated using the algorithm of Gascoin et al., (2015).





## 3. METHOD

### 3.1 SnowModel description

The physically based model SnowModel (Liston and Elder, 2006b) was used to simulate the snow depth evolution over the entire catchment. SnowModel has already shown acceptable performance in the challenging context of semi-arid mountains, including the Andes (Gascoin *et al.* 2013, Mernild *et al.* 2017) and the High Atlas (Baba et al., 2018a, 2018b). It is a spatially distributed snowpack evolution modeling system composed of four submodels briefly described below.

#### a/ MicroMet

Micromet is a physically-based meteorological distribution model developed specifically to produce high-resolution, spatially-distributed atmospheric forcing data. This model requires precipitation, wind speed and direction, temperature and humidity as input data, generally measured at weather stations. For the incoming solar and longwave radiation, and surface pressure, MicroMet can either compute these fields from other meteorological variables, or create them from observations through a data assimilation procedure (Liston and Elder, 2006a). MicroMet includes a preprocessor component that first analyzes meteorological data to identify and correct potential deficiencies (e.g. values out of the ranges given in the subroutine). It then fills in any missing data segments with realistic values. The atmospheric fields are distributed using a combination of lapse rates and spatial interpolation using the Barnes objective analysis scheme (Barnes, 1964).

#### b/ EnBal

EnBal performs standard surface energy balance calculations (Liston, 1995; Liston et al., 1999). This component simulates surface temperatures and energy fluxes in response to observed/modeled near-surface atmospheric conditions provided by MicroMet. Surface latent and sensible heat flux and snowmelt calculations are made using a surface energy balance model.

#### c/ SnowPack

SnowPack is a single or multi-layer (max. 6 layers), snowpack evolution and runoff/retention model that describes snowpack changes in response to precipitation and melt fluxes defined by MicroMet and EnBal (Liston and Hall, 1995; Liston and Elder, 2006b).



### d/ SnowTrans-3D

SnowTran-3D (Liston and Sturm, 1998; Liston et al., 2007) is a three-dimensional model that simulates snow depth evolution (deposition and erosion) resulting from windblown snow based on a mass-balance equation that describes the temporal variation of snow depth at each grid cell within the simulation domain.

## 3.2 Model set up

### 3.2.1 Spatialized meteorological forcing

Spatial interpolation using the Barnes scheme was used to distribute the nine AWS measurements of Tair, RH, LW, SW and pressure over the model domain. Since the relative humidity is a non-linear function of elevation,
the relatively linear dewpoint temperature is used for the elevation adjustment. Fore more details please refer to Liston and Elder 2006. In this study the MicroMet subroutine has been run with the default setting for the south hemisphere, for air temperature and dewpoint temperature monthly lapse rates (Liston and Elder, 2006b). Monthly lapse rates computed from the available measurements are dependant on the year considered. As the mean is close to the default settings, it has been chosen to conserve these values. The model has been run on the
SRTM DEM and as a result, hourly meteorological data over a 100m-grid resolution are available for entire study period. Precipitation was interpolated similarly but without considering a lapse rate, as the comparison between the available measurements did not reveal consistent elevation gradients. Wind data and direction were first interpolated using linear lapse rates and then each gridded value was corrected considering topographic slope and curvature relationships (Liston and Elder, 2006b).
The 3km WRF outputs (section 2.2.1) were inputs for MicroMet which considers that each WRF cell corresponds to a virtual weather station located in the center of the WRF cell, following Mernild et al. (2017) and Baba et al. (2018a). MicroMet adjusts the elevation bias to the DEM at the corresponding coordinate and downscales the data to a 100 m grid.

### 3.2.2 Albedo calibration

The snow albedo evolution is computed as a function of the snow density and air temperature (more details in Liston and Hall, 1995; Liston and Elder 2006b). Minimum and maximum values have been adjusted based on measurements. The minimum (*i.e.* the soil) is fixed at 0.2 and is quite homogeneous in this basin, as there is almost no vegetation. The minimum and maximum snow albedo (corresponding to old and fresh snow,
respectively) are respectively fixed to 0.6 and 0.9 in agreements with measurements performed at the AWS (Figure 2).



### 3.2.3 Turbulent fluxes calibration

As the model is using a bulk approach to simulate the turbulent fluxes, the turbulent latent and sensible heat fluxes (respectively *LE* and *H)* are parameterized using an effective surface roughness length $z_0$ (Liston, 1995;

Liston et al., 1999). Note that this roughness length $z_0$ is considered as an effective value used in the model to represent the aerodynamic ($z_m$), temperature ($z_t$) and humidity ($z_q$) roughness values. As no measurements are available to calibrate/validate this value, it was initially fixed at 1mm (Gromke et al., 2011; Shelley MacDonell et al., 2013), and a subsequent sensitivity test was undertaken.

### 3.2.4 Wind transport parameterization

The model considers the wind transport (saltation, turbulent suspension) after snow deposition, sublimation of blowing and drifting snow and erosion and deposition after snowfall, depending on the topography (Liston and Sturm 1998). The topographic influence on wind transport has been set, following Gascoin et al., 2013. The curvature allows considering the typical redistribution length scale. Based on the DEM, it was estimated to be

500 m, *i.e.* approximately one-half the wavelength of the topographic features within the domain (Liston et al., 2007). The model considers different weights for slope and curvature and values. We have chosen 0.58 and 0.42, respectively, following Gascoin et al., 2013.

### 3.3 Simulations

Two types of simulations have been performed over the entire catchment for the period 1[st] of January 2014 – 1[st] of January 2016 on a 100-m resolution DEM. The first simulation was forced with input from the nine automatic weather station measurements (referred to as AWS-forcings), whereas the second simulation was forced with the WRF data (referred to as WRF-forcings).

After indicating the differences observed for these two forcing, the model is primarily validated at local points.

Results for the two simulations were first compared to local snow depth measurements at each AWS (described in section 2.2.1). The performance was evaluated using a Kappa statistic coefficient (Cohen, 1960) denoted *k,* to measure the agreement between the simulation and the observation, considering the percentage of time with and without snow. The Root Mean Square Error (*RMSE*) was also calculated.

Second, the model performance was evaluated over the entire catchment, by comparing the simulated snow cover

extent and duration to that observed by the satellite images (described in section 2.2.2). The model performance



was evaluated by computing the Nash-Sutcliffe efficiency coefficient (*NSE*, Nash and Sutcliffe, 1970) between simulations and observations, and the *RMSE*.

After validating the model, the sublimation ratio and rate were computed over the catchment for the two years. The sublimation rate corresponds to the mass sublimated per unit of time. The sublimation ratio is defined as a percentage, and equal to the sublimation divided by the total ablation (i.e. sublimation and melt rates).

Finally, additional simulations were performed over the same study period to evaluate and discuss the sensitivity of the model to the main uncertainties. First a sensitivity test for precipitation uncertainties was performed by increasing the AWS precipitation data by 10%. Second, sensitivity tests to model calibration were completed by *(i)* increasing the roughness value by a factor 10, *(ii)* changing the curvature length with values ranging between 100 m to 1000 m and *(iii)* changing the relative influence of slope and curvature between 0.25 to 0.75.

## 4. RESULTS

### 4.1 Meteorological forcing comparison

### 4.1.1 2014 *vs* 2015

According to the AWS measurements, the beginning of the year 2015 (i.e. from January to July) was warmer than 2014. Conversely, for the second part of the year (from June to December), observations indicate lower temperatures for 2015 than for 2014 (mean difference of -2.6°C). Higher values of relative humidity were observed for the entire 2015 period compared to 2014 (mean difference of 11%).

Lower incoming shortwave (mean difference of -18 W.m$^{-2}$), with larger differences over the July-December period (mean difference of -32 W.m$^{-2}$), were observed for 2015. The lower incoming SW radiation observed for the second part of 2015 is in agreement with lower air temperature. In addition, lower incoming SW is explained by a larger number of clouds during this period in the catchment for 2015. Concurrently, larger incoming longwave radiation was also observed for 2015 (mean difference of 20 W.m$^{-2}$), consistent with an increased cloudiness that year.

### 4.1.2 AWS *vs* WRF

### a/ At the weather station

A comparison between each weather station and the closest WRF virtual station was completed. A special offset between the weather station and the WRF point of a few meters can exist, but this is small compare to the WRF elevation bias and therefore not considered. The most important differences between AWS and WRF-forcings are for the air temperature, the relative humidity and the precipitations rates.





When comparing WRF 2-m air temperature with AWS measurements, results indicate a cold bias in WRF outputs. While the correlations are in good agreement ($0.91 < R < 0.97$ for the nine stations), the *RMSEs* indicate a significant and systematic bias ($5°C < RMSE < 7°C$), with larger differences observed at higher elevations (i.e. Paso Aguas Negras, Vega Tapado). Significant biases in RH are also observed, with lower WRF-RH compared to

the AWS measurements (*RMSE* ~20 %).

Precipitation rates from WRF are usually larger than measured rates for each event (mean difference at La Laguna of 3.7 mm.d$^{-1}$ in 2014, 13.1 mm.d$^{-1}$ in 2015, 2.9 mm.d$^{-1}$ in 2014, and 9.1 mm.d$^{-1}$ in 2015 at Tapado). In addition, there are more precipitation events for WRF than measured in the catchment (6 more days in 2014 and 48 more for 2015 at La Laguna). Additional WRF precipitation events compared to observations are mainly

related to drizzle effects. As a result, WRF reports a larger amount of annual cumulative precipitation. For instance at La Laguna AWS, the annual measured cumulative precipitation was 70 mm w.e. for 2014 and 440 mm w.e. for 2015. WRF simulations indicate an annual cumulative precipitation of 320 mm w.e. for 2014 (i.e. 4.3 times larger than the observations) and 760 mm w.e. for 2015 (i.e 1.7 times larger). Differences are even larger at higher elevations. At Tapado, annual cumulative precipitation was 100 mm in 2014 and 460 mm w.e. in

2015, while WRF precipitation was 680 mm w.e. for 2014 and 1330 mm w.e. for 2015. Here, the WRF precipitation was 7 and 2.9 times larger than measured values for 2014 and 2015, respectively. There is no general trend observed over space or time for the annual cumulative precipitation as it strongly depends on the year, the month, and may be unique for each event.

**b/ MicroMet output comparison**

The MicroMet outputs differences, when using WRF versus AWS inputs indicate similar results to station comparisons (Figure 4). The annual mean temperature differences range from 4.5 to 7.5 °C depending on the year and the elevation, and confirm colder temperature when using WRF-forcings. Similar results are found for RH (differences between 13 and 24%, with lower values when using WRF-forcings). Finally WRF simulates more

precipitation (Figure 4d), with the maximum difference (annul cumulative difference larger than 1 m w.e.) at higher elevations. Despite these significant differences, both forcings were used as inputs in order to quantify the impact of the forcing choice on the sublimation estimation in this study.





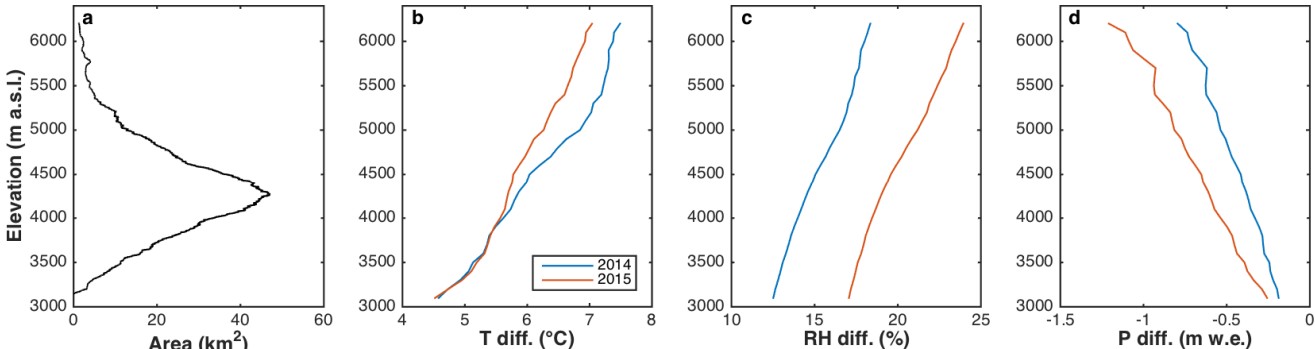

*Figure 4: (a) Area-elevation distribution of the studied catchment. Mean difference at the catchment scale for 2014 and 2015 between MicroMet outputs forced by the AWS and the WRF data for (b) temperature, (c) relative humidity and (d) annual cumulative precipitation.*

## 4.2 Model performances and validation

### 4.2.1 Comparison with local snow depth measurements

Simulations performed using the AWS-forcings (Figures 5 a-f) are in good agreement with measurements (mean $k$=0.14 and mean *RMSE*=0.15 m). While comparisons have been performed at distinct stations for 2014 and

10   2015, we can observe better performances (i.e. higher $k$ and lower *RMSE*) for 2015 (Figures 5 d-f) than for 2014 (Figures 5 a-c). For 2014 the highest $k$ and lower *RMSE* are observed at Tapado AWS, as precipitation measurements were available at this site, but performances are much lower at the two other sites where precipitation was interpolated. In 2015, the simulated snow depth is generally overestimated, probably due to an over-estimation of the precipitation for the large event in June (Figure 3). Nevertheless, the starting date and last

15   day of snow are in good agreement with observations (maximum difference of 3 days observed at La Gloria site). Note that this comparison probably over-estimates the accuracy as snow depths are compared at the exact location of meteorological forcing. Larger uncertainties are expected at the interpolated locations.

Simulations performed with the WRF-forcings indicate lower performances in simulating snow depth evolution at the AWS (Figure 5 g-m; mean $k$=0.12, and mean *RMSE*=0.20 m). The results indicate an over-estimation of

20   the simulated snow depth compared to the observations. In addition, for 2014, the timing of the first day of snow and the day of the snow disappearance do not fit well with observations (and explain the low $k$ values). In 2015, the first day of snow is generally in good agreement with observations (maximum difference of 5 days observed at La Laguna).



While AWS-forcings indicates better performances, in both cases, better results are found for the year 2015. The dry condition of 2014 can explain the complexity in considering spatial variability of the precipitation. Due to very low snow amount there is probably local snow patches, complex to represent in models. Therefore, it can explain the lower performances of the model in simulating the snow depth evolution for this year.

5 These results also underline the complexity in modeling the spatial variability of SD, even though snow transport is considered. Results show an overall similarity of the simulated SD between some stations (e.g. Vega Tapado, Colorado Alto and La Gloria), while measurements indicate that SD are much more variable in reality. Note that the windy conditions on the local depression at Vega Tapado is very local (i.e. few meters) and make complicated the simulation at this site where the measured SD is larger that the surrounded area and not representative of the
10 100m grid cell.





*Figure 5: Simulated (cyan/purple) vs. observed (green) snow depth at 6 automatic weather stations. Blue lines (a-f) represent simulations performed using AWS meteorological forcing. Purple lines (g-m) correspond to simulations performed using WRF meteorological forcing. Green shaded areas indicate the period of available measurements.*



### 4.2.2 Comparison with satellite images

The snow cover area (SCA) and the snow cover duration (SCD) over the entire catchment are compared to the MODIS product. Note that this 8-days product can overestimate the snow cover since only one day of snow cover

is required to classify all 8 days as snow covered.

**a/ Snow cover area**

The simulated snow cover area (SCA), forced by the AWS-forcings is in good agreement with observations from MODIS products (Figure 6a) with, in particular, a good simulation of the timing of precipitation events. Best fits are observed for the winter and spring 2015 (*i.e.* from July to December), with the higher correlations

($NSE_{SCA}$=0.94, $RMSE_{SCA}$=41.6 km$^2$ (i.e. 8.3%)). Regarding the ablation, in June 2014 and April - May 2015, the simulated SCA decreases faster that the observed SCA, which can be due to an over-estimation of melt or/and sublimation or an underestimation of the accumulation.

When using the WRF-forcing, the agreement between SCA to MODIS is relatively poor (Figure 6b). The timing of snowfall events is not always in good agreement with the observations due to missing events (e.g. March

2015), a timing bias of a few days (e.g. March 2014) and/or additional events (during both 2014 and 2015 winter season). The simulated SCA evolution over winter and spring of 2015 shows strong variations over the entire catchment, which are not observed in the MODIS record.

Here again, for both forcing datasets, better performances are observed for 2015 ($NSE_{AWS}$ = 0.79, $RMSE_{AWS}$ = 93.2 km$^2$ (i.e. ~19% of the total area); $NSE_{WRF}$ = 0.61, $RMSE_{WRF}$ = 125 km$^2$ (~25%)) than for 2014 ($NSE_{AWS}$=0.41,

$RMSE_{AWS}$=117 km$^2$ (~23%) ; $NSE_{WRF}$ = 0.23, $RMSE_{WRF}$ = 133 km$^2$ (~27%)).





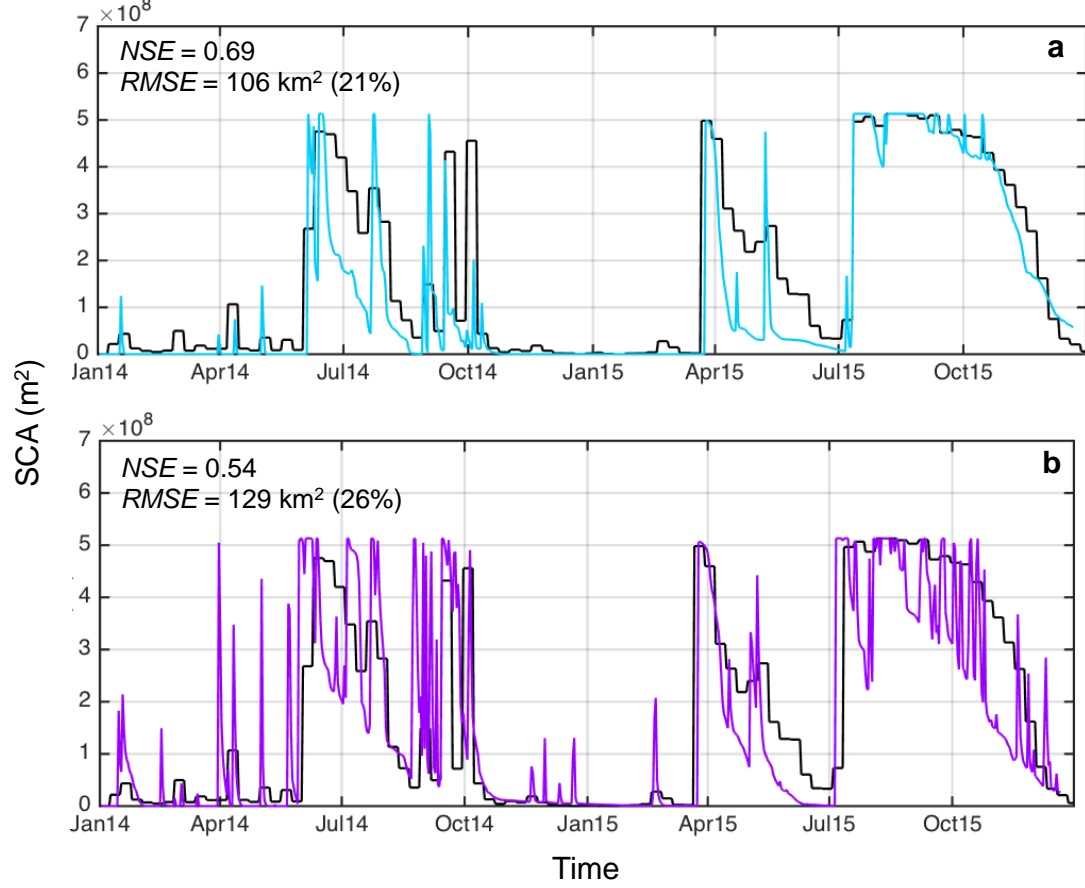

*Figure 6: Snow cover area evolutions over the 2014-2015 period, from MODIS images (black lines), and simulations using (a) AWS-forcing (blue line) and (b) WRF-forcing (purple line).*

### b/ Snow cover duration over the catchment

The simulated snow cover duration (SCD) was also compared to the observed duration (from MODIS) for each 200 m elevation band. Better performances were obtained for the AWS-forcings than for the WRF-forcings, for both years (Figure 7).

Results based on the AWS-forcings are in good agreement with observations at low elevations (i.e. below 4600 m a.s.l.; Figure 7), but show an over-estimation of the SCD at high elevation (absolute mean difference of 30 and 27 days for 2014 and 2015 respectively).



When using the WRF forcing, SCD is over-estimated for the entire catchment in 2014 (absolute mean difference of 67 days). In 2015, simulations indicate an over-estimation of the SCD at low elevation (i.e. below 4500 m a.s.l.), and a small under-estimation at higher elevations (absolute mean error of 34 days for 2015).

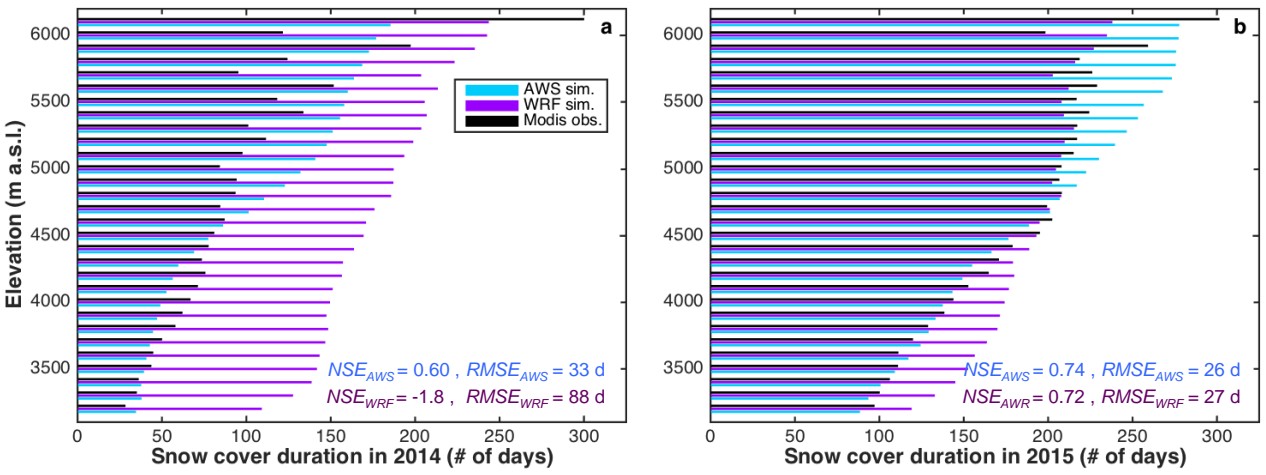

*Figure 7: Snow cover duration per each 200m elevation band, from MODIS images (black), AWS-forcing (blue) and WRF-forcing (purple) for (a) 2014 and (b) 2015.*

## 4.3 Energy balance results

Energy balance fluxes were averaged over snow surfaces (Figure 8). Results indicate that except for the AWS-simulation in 2014, turbulent fluxes ($QE$ and $QH$) are the main energy fluxes contribution. The net SW is a significant contribution, and relatively similar for all the simulations. For the annual mean AWS-simulation net SW is 18 and 22 W.m$^{-2}$ for 2014 and 2015, respectively. For the mean AWS-simulation net SW is 24 and 23 W.m$^{-2}$ for 2014 and 2015, respectively. For all other scenarios, the net LW is low for all simulations (annual mean of 6 and 7 W.m$^{-2}$ for WRF and AWS-simulations respectively), due to the very dry conditions of the atmosphere.

Significant differences are observed depending on the forcing used. For 2014 (Figure 8 a,c), using the WRF-forcings, there is a larger contribution of the turbulent fluxes which become the main contribution. This significant difference can be explained by the larger snow cover simulated by WRF-forcings. Indeed, the snow cover simulated when using the WRF-forcings generally covers the entire catchment while the AWS-forcings leads to snow over the upper part of the catchment only. Therefore, as the fluxes are computed over the snow surface, WRF increases the contribution of warmer, low elevation areas. In both cases, the annual mean Bowen



Ratio ($Br = Qh/Qe$) is larger than one ($Br$ = 2.3 and 1.2 when using the AWS, WRF-forcings respectively),
indicating that a greater proportion of the available energy at the surface is passed to the atmosphere as sensible
heat than as latent heat.

While lower differences are observed for 2015 (Figures 8 b,d), the $Br$ strongly depends on the forcing used: 1.9

5   for the AWS-forcings and 0.9 when using the WRF-forcings. This means that in one case there is a greater
proportion of the available energy at the surface that is passed to the atmosphere as sensible heat than as latent
heat, while the opposite occurs for the second case.

Despite the difference between forcing, in both cases, larger $Br$ are found for 2014 than 2015.

Nevertheless, part of the differences between years and forcings seem to be related to the snow depth differences,

10   an aspect further discussed in section 5.

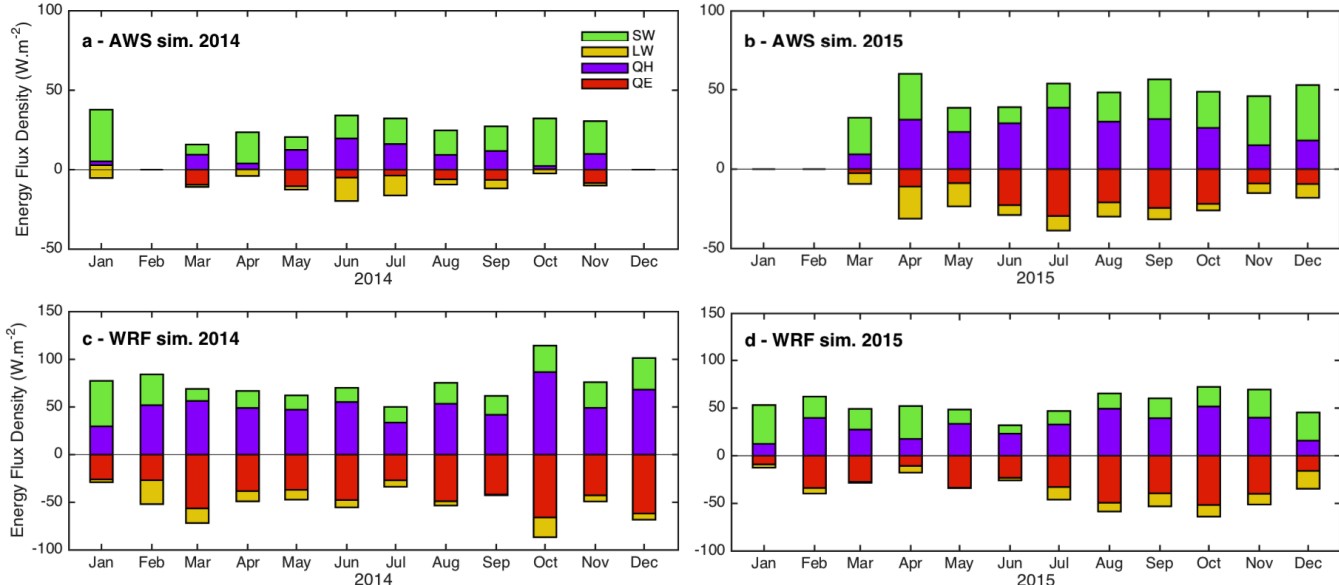

*Figure 8: Monthly average of the main modeled heat fluxes for the entire catchment, over snow surfaces only.
SW is net shortwave radiation, LW is net longwave radiation, QE is the latent heat flux and QH is the sensible*

15   *heat flux.*





## 4.4 Simulated sublimation

### 4.4.1 AWS *vs* WRF forcing

The choice of the forcing used strongly impacts the simulated sublimation ratio (Figure 9 a,b). The annual means computed over the entire catchment are 42% and 49% for 2014 and 2015, respectively, when the model is forced with the AWS-forcings (Figures 9 a,b). It reaches 86% (2014) and 80% (2015) when the WRF-forcings are used (Figures 9 c,d). These values are highly variable over the year, but for both forcings, larger sublimation ratio and rates are generally observed during the winter season (i.e. from July to October), when the snow is covering a larger part of the catchment.

The sublimation ratio is also variable in space. For both years and both forcing results, the sublimation ratio increases with elevation. Regarding the comparison between the forcings, larger discrepancies are observed at lower elevation, especially below 5300 m a.s.l. (~50% of differences for 2014 and ~30% for 2015). Note that larger differences were observed for 2014 related to larger differences in snow cover and snow duration between forcings.

Regarding the simulations performed with the AWS-forcings, melt predominates at all elevations (Figure 10 c,d). In addition, both melt and sublimation rates increase with elevation until 5300 m a.s.l.. Above this elevation the melt rate is relatively constant or even decreases, explaining the larger values of the sublimation ratio at higher elevation.

Regarding the simulation performed with the WRF-forcings, sublimation rates are larger than the melt rates at all elevations. Above 3800 m a.s.l. melt is relatively constant with elevation and the sublimation ratio strongly increases, explaining the larger values of the sublimation ratio at high elevations.





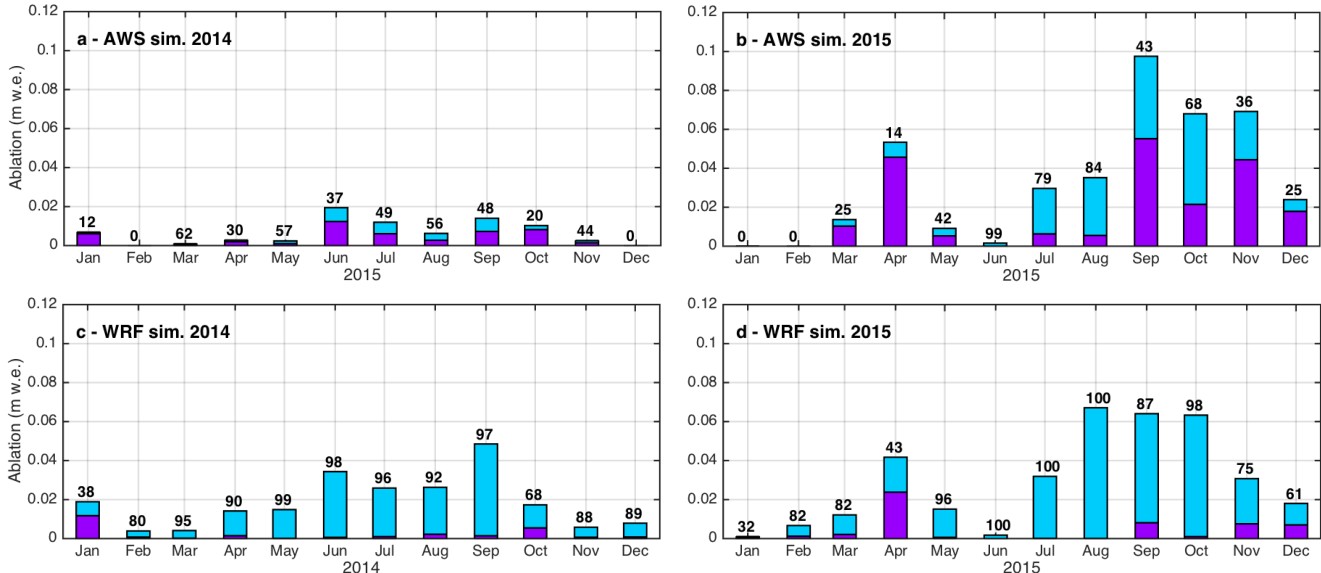

*Figure 9: Monthly simulated melt (purple) and sublimation (blue) using AWS-forcing (a,b) and WRF-forcing (c,d). Black numbers indicate the monthly sublimation ratio in %.*







*Figure 10: Simulated annual sublimation ratio (a,b) for each 200 m elevation band using AWS (blue) and WRF (purple) forcing for 2014 (a,c) and 2015 (b,d). Simulated annual ablation ratio (sublimation and melt) against the elevation using AWS and WRF-forcing for 2014 (c) and 2015 (d).*

## 4.5 Sensitivity tests on the AWS-forcings simulations

Four different parameters were tested to evaluate the uncertainties related to the calibration of modeled parameters: roughness value, precipitation amounts (due to measurement uncertainties), topographic curvature length and slope versus curvature length (Figure 11; Supplementary Information B). The results showed that the sublimation ratio was most sensitive to roughness values, and that differences due to the other three variables were an order of magnitude lower. For more details, please refer to the supplementary information.





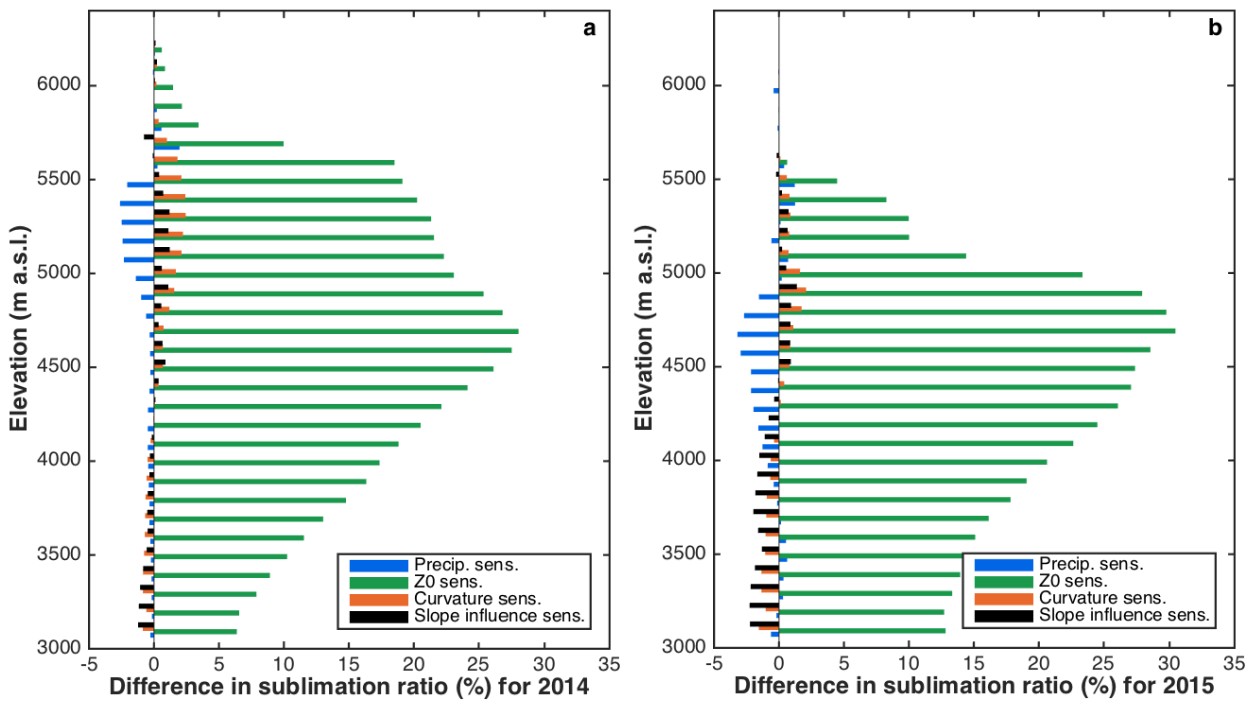

*Figure 11: Differences in sublimation ratio per elevation ranges for (a) 2014 and (b) 2015, between: reference simulation and the simulation performed with a precipitation increase at the two AWS by 10% (blue); reference simulation ($z_0$=1mm) and with an increase of $z_0$ to 10mm (green); 100 and 1000m curvature length simulations (orange) and slope vs. curvature weight of 0.25 – 0.75 and 0.75 – 0.25 (black).*

## 5. DISCUSSION

### 5.1 AWS *vs.* WRF forcing.

#### 5.1.1 Air temperature

A cold bias in air temperature, using the combination NCEP-WRF, is often observed and well documented (e.g. Ruiz et al., 2010). It can be explained by the model parameterization complexity such as: *(i)* the initial or lateral conditions especially for the land surface surface temperature (Cheng and Steenburgh, 2005) or soil thermal conductivity (Massey et al., 2014), *(ii)* the parameterization of the planetary boundary layer scheme (Reeves et al., 2011), or *(iii)* the radiation parameterization scheme as it has been observed for other models (e.g. Müller and Scherer, 2005). However, the exact source of this bias remains difficult to identify (e.g. Reeve *et al.,* 2005). In



this study, the default parameterization has been used, but works are in progress regarding the evaluation of the most appropriate calibration over this area, using direct observations.

### 5.1.2 Precipitation

Precipitation from the WRF model is known to be over-estimated, particularly in the Andes (e.g. Mourre et al., 2016). One possible explanation is that biases can exit in the reanalysis data, in particular at high elevations, where observations are often scarce. Precipitation over-estimation might also be related to the parameterization used for the model which may not be the most appropriate for the Andes; further work is needed to determine the most appropriate ones. Outputs may also be inaccurate due to the relatively low-resolution DEM used (100 m).

AWS measurements may also be biased since snowfall measurements caught by each rain gauge can be underestimated, due to the wind (e.g. MacDonald and Pomeroy, 2007; Wolff et al., 2015). This gauge undercatch uncertainty (see section 5.4.1) could increase the difference between the precipitation simulated by WRF and that measured at the AWSs. In addition, questions arise regarding the representatively of the punctual measurements compared to the grid cell considered in the model.

The spatio-temporal variability observed in the difference between AWS and WRF precipitation data, highlight that it would be inappropriate to use a constant correction factor to adjust WRF data with measurements. More studies comparing WRF output to AWS among other data sets are required to determine a realistic correction method.

### 5.1.3 Impact on the sublimation

Lower temperature and relative humidity values from WRF outputs compared to AWS measurements can explain, in part, the larger simulated sublimation ratio found with this forcing (Figure 9 and 10 a,b). The relatively high amounts of precipitation simulated by the WRF outputs, and resulting snow cover overestimation, may also play a role. Differences in the sublimation ratio when using the AWS and WRF-forcing are quite similar for the two years (mean annual difference of 42% and 36% for 2014 and 2015 respectively), although the

difference between the melt rate and sublimation rate depends on the year (Figure 10 c,d).

Larger differences in sublimation rates between the two forcing datasets are observed in 2014, especially at high elevation. During this year there are lower RH and P biases (Figure 4c,d), but larger air T differences, especially at high elevations (Figure 4b). On the other hand biases in wind speed, incoming LW and SW and air pressure are low for the two years (results not shown).

The larger RH bias in 2015 indicates an over-estimation of the dryness for this year (compared to 2014), and would lead to larger differences in sublimation rate for 2015 than for 2014, which is the opposite of the results



observed. Likewise, the larger over-estimation of precipitation amount observed in 2015 (Figure 4d) does not explain the larger difference in sublimation, as a deeper snow depth should result in more persistent snow cover during the warm period and hence a lower sublimation ratio related to larger melt rate. Therefore, although the relationship between temperature and sublimation rate is complex and not necessary direct, in this case, the

colder temperature is the most probable explanation for at least part of the larger difference in sublimation rate observed at high elevation.

The SCD may also have a significant influence on sublimation. For 2014, the differences in SCD between the two forcings were ~100 days below 4500m, and close to 50 days above this elevation (Figure 7a). Snow that persists over a longer period results in an increased ablation rate and can influence the sublimation rate and ratio.

This is the only explanation for the larger melt rate observed with AWS-forcings compared to WRF-forcings at low elevations, given the cold bias in WRF-forcings (Figure 10c and 4b). Since a larger SCD can be related to larger precipitation amount, precipitation uncertainties likely play a significant role and the sublimation estimation.

In summary, these results indicate that the simulation performed with AWS-forcings leads to better performance

than when using WRF forcing. The following discussion is thus based on the AWS-forced results.

## 5.2 ENSO events (2014 *vs.* 2015) and sublimation ratio

In this region, 2015 was considered as a strong *El Nino* event, associated with warm and wet conditions. On the contrary 2014 was dryer and colder and considered a neutral year (Ceazamet; https://iri.columbia.edu). We would

therefore expect a larger sublimation ratio for the winter of 2014.

The winter was dryer in 2014 compared to 2015 according to meteorological measurements made in the region. Temperature measurements from the AWS in the catchment (see section 4.1.1) are in agreement with these general observations for the first part of the year (i.e. from January to July), but for the remainder of the year AWS measurements are colder in 2015 compared to 2014. The compensation between a dry year (with low snow

amount) and cold spring and summer, *versus* a wet year with warmer spring and summer can explain the similarity between the sublimation ratios observed in 2014 and 2015 (42 *vs.* 49%).

Although the sublimation ratios are similar for both years, there is a significant sublimation rate over the long melting period in 2015. This may be explained by warmer conditions which induce a warmer snow pack, increasing the saturated vapor pressure at the snow surface, providing energy to increase the sublimation rates.

This process is called evaposublimation (Herrero and Polo, 2016).





### 5.3 SD influence on the sublimation ratio

### 5.3.1 Energy balance with precipitation held constant

As mentioned in sections 5.1.3, snow cover has a significant influence on sublimation. Thus, in order to compare the energy balance differences between years, we held the precipitation input constant. We applied the 2014 precipitation amount to both years, then completed a second simulation using the 2015 precipitation for both years.

**a/ Difference between the two years**

The differences between 2014 and 2015, using 2014 precipitation (i.e. dry conditions), indicate a difference in sublimation ratio of 2% (i.e. 42% for 2014 and 44% for 2015). While the ratio is quite similar, the sublimation rate is larger for 2015 from June to November (Figure S2 a,c, in the Supplementary Information). This may be related to increased wind speed recorded in 2015 which would result in a larger contribution of turbulent fluxes and therefore sublimation (Figure 12 a,c). During the same period in 2014 melt predominates (Figure S2 a), related to warmer temperatures for the second period of 2014.

Results are similar when using the 2015 precipitation data (i.e. wet conditions), but there are larger differences between years. Indeed, the sublimation ratio difference is 17% (i.e. 30% for 2014 and 47% for 2015). The energy contribution is very distinct. For example, in 2014 *SW* is the main energy flux contributor, and the contribution of the *LW* is greater in 2014 than 2015 (probably due to lower temperature). In 2015 turbulent fluxes (*QH* and *QE)* have a larger contribution than the *SW* (Figure 12 b,d), related to lower temperature and larger wind speeds. For this simulation, the increased sublimation in 2015 is likely due to the increased importance of turbulent fluxes as well as a lower melt responsible for a longer snow persistence on the ground (i.e. until December in 2015, and October for 2014) (Figure S2 b,d, in the Supplementary Information). Note that for wetter conditions (i.e. larger snow amount), the difference in sublimation ratio is much larger, related to a larger area covered by snow over a longer time period.

**b/ Impact of the snow cover**

For exactly the same meteorological conditions, increasing the precipitation amount (i.e. using 2015 precipitation data) reduces the sublimation ratio by 12% for 2014 and by 3% for 2015. In term of energy balance we can see a larger contribution of the turbulent fluxes for the wet conditions (Figure 12c) than for the dry condition (Figure 12a). Similar observations are found for 2015 (Figure 12b,d), but with lower differences. These results indicate an increased sublimation ratio related to a decrease in snow depth, with increased sensitivity for the dry year due to the influence of the SCD and SCA.





## c/ Conclusion

Differences in terms of sublimation ratio are observed, depending on the forcing condition (i.e. 2014 versus 2015 precipitation amounts applied to both years). These results highlight the importance of snow depth on the sublimation, as increased snow depth leads a lower sublimation rate mainly due to snow cover persistence over the warm season, which favors melt. In addition, larger differences are observed for the 2014 forcing condition, highlighting the importance of the snow cover area and duration over the catchment. For example, when the mean sublimation rate or ratio is computed over SCA, there is a larger difference in the observed sublimation between years since sublimation is dependent on elevation and the SCA distribution with elevation is more variable for the 2014 forcing.

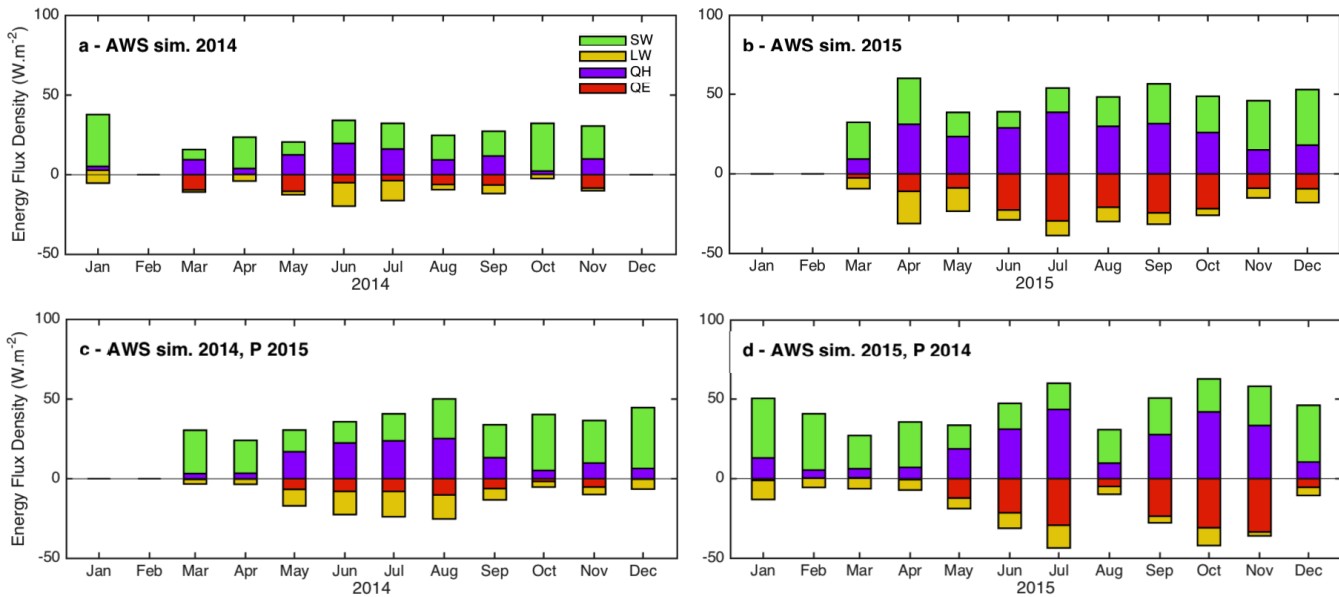

*Figure 12: Monthly average (for the entire catchment, over snow surfaces only) of the main modeled heat fluxes for (a) 2014 and (b) 2015 AWS-forcings, (c) a simulation using the 2015 precipitation and the 2014 AWS-forcings, and (d) a simulation using the 2014 precipitation and 2015 AWS-forcings. SW is net shortwave radiation, LW is net longwave radiation, QE is the latent heat flux and QH is the sensible heat flux.*

## 6. CONCLUSION

In this study, the snow cover and sublimation has been simulated over La Laguna catchment located in the semi-arid Andes of Chile. Using the snow pack model SnowModel, simulations were performed over two contrasting



years (2014 considered as a dry year and 2015 considered as a wet year), using two distinct forcings (nine AWS located in the catchment and 3km resolution WRF model outputs).

Results indicate strong differences in simulated snow depth depending on the forcing chosen, mainly due to a cold bias in air temperature in WRF as well as an over-estimation of precipitation. As a result, performances in
simulating the snow cover using the AWS-forcings are more realistic, both at the local scale (comparison with AWS snow depth measurements), and over the entire catchment (comparison with SCA and SCD from MODIS images). In addition, independently of the forcing choice, the simulation of snow cover is better for 2015 compared to 2014, mainly due a larger sensitivity to the precipitation uncertainties during dry conditions. This highlights the complexity in properly modeling the snow cover evolution for years with low precipitation years.
There are also large differences in modeled sublimation ratio depending on the forcing chosen. When using WRF-forcings the sublimation ratio is approximately twice that modeled with the AWS-forcing. This is partially due to the differences in temperature and relative humidity between the two forcings, but mostly due to precipitation differences. For example, when holding all model inputs constant except for precipitation, there are significant differences in the modeled sublimation, especially for 2014 which was a dry year. Otherwise, the
annual mean of the sublimation ratio over a catchment is similar during the two years and but it increases with elevation. This partly explains the larger sublimation values reported in previous studies performed at high elevations in the semi-arid Andes of Chile (e.g. Gascoin et al., 2013; Ginot et al., 2001; S. MacDonell et al., 2013).

Sublimation simulated in this study is associated to several sources of uncertainty. First, the precipitation data is
the main forcing uncertainty due to measurement errors and lack of spatial representation as precipitation data was only available for two stations. This study highlights that this uncertainty has a strong impact on sublimation and further work is suggested to *(i)* improve measurements uncertainties, *(ii)* increase the number of sensors over the catchment, and *(iii)* incorporate AWS measurements into the WRF model and use data assimilation to improve model outputs. The CEAZAmet group is currently working on point *(iii)* to provide improved WRF
outputs for the semi-arid Andes of Chile.

The main uncertainty related to model calibration is due to the absence of turbulent flux measurements needed to properly calibrate the roughness value. Eddy covariance measurements were initialized in 2017 at the Tapado station and will provide useful data to calibrate this value and better evaluate the uncertainty.



**Acknowledgment**

The authors thank G.E. Liston for providing SnowModel code and for the interesting and useful discussions and suggestions. We are also greatful to thank P. Salinas for providing the WRF simulation outputs and for providing a useful help related to these data. C. Kinnard was supported by a *Coopération bilatérale – Québec-Chili*
*Ministère des Relations Internationales et Francophonie.*

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
