# Peer review of "Uncertainties in the spatial distribution of snow sublimation in the semi-arid Andes of Chile"

_The Cryosphere, 2019_

## Referee Comment (RC1) · Anonymous Referee #1 · 28 Mar 2019

(A) General Comments

In the manuscript "Uncertainties in the spatial distribution of snow sublimation in the semi-arid Andes of Chile" Réveillet et al. present results of their study aiming to simulate melt and sublimation rates over the instrumented watershed of La Laguna. They present the relative importance of sublimation versus snow melt using a distributed snowpack model for two meteorologically contrasting years. They detected a large difference in modelled sublimation rates forcing the model with data of Automatic Weather Stations (AWS) and with Weather Research and Forecasting (WRF) model data. This difference is caused by (a) the different meteorological input, particularly in precipitation and temperature, and (b) by the modelled snow cover persistence.

The objective of the study of Réveillet et al. is to assess the uncertainties in melt

and sublimation arising from modelling snow evolution using AWS and/or WRF-model generated meteorological datasets. Since the seasonal snow cover and glacier melt as processes of the cryosphere have locally a high contribution to available fresh water in this region. The study of Réveillet et al. contributes to gain knowledge of snow depth distribution and snow cover processes in the semi-arid Andes of Chile, and thus the manuscript is in the scope of TC. In general, this study shows again the importance of a critical interpretation of model results with respect to model input data.

The introduction is complete, the applied methods are appropriate and comprehensible, and the results are compared to referenced work. However, the manuscript at its present stage summarizes results of interesting work packages and analysis, but lacks in the overarching aim with a clear problem statement, the research questions and the respective conclusions. This is also obvious in the high number of subsections presenting methods and results which not necessarily contribute to the conclusions. My suggestion is to restructure the manuscript with a clear focus on formulated research questions and to revise the title reconsidering the term "uncertainty".

(B) Specific Comments

(1) As addressed in the general comments, the manuscript shows an interesting work but without presenting an overarching aim and respective research questions. Forcing the model with different data from AWS and WRF does not result only in differences of sublimation rates, but also differences in e.g. snow covered area, snow persistence and snow melt. So I would suggest that not the uncertainties in the spatial distribution of snow sublimation are shown. Rather the variation of snow related parameters and processes forcing the applied model with different input is presented. Since the weaknesses of the WRF model output are known (cold bias, precipitation overestimation) and the AWS data might be the more appropriate model input, the study shows the error of forcing the model with WRF data for snow parameters (i.e. the overestimation of the snow cover duration by 2 month). The impacts of these errors are particularly obvious in the sublimation rates and ratios, which again are a function of snow coverage, elevation, temperature, etc. If the (model?!) uncertainty in sublimation should be addressed in more detail, I would suggest to show at least one additional figure presenting the calculated and simulated sublimation rates at AWS locations. In my opinion, the manuscript at its present stage presents "Differences in simulated sublimation in a high mountain catchment of the semi-arid Andes of Chile using AWS data and WRF meteorological forcing". Possible research question may address the main differences (errors?) in sublimation simulated using the WRF forcing, the impact of SCA and SCD under/overestimation, and the effect of different meteorological conditions of the two contrasting years on sublimation ratios. This can lead to conclusions about advantages and disadvantages of using the different forcing data. Most of the results are already presented, but the paper should be restructured accordingly to answer the formulated research questions and further drawing the conclusions.

(2) Two years with contrasting meteorological conditions have been chosen for this study. This has the advantage of testing the simulation results, but also seems to be restricted to AWS data availability. To get an overview of the overall climate in this region and to classify the two selected years in a climatological context, please present a short climate overview of the last 30 years from a nearby station, or at least some short statistics for the AWS with the longest data history (La Laguna?)

(3) The downscaling of the WRF data still appears opaque to me. Is the relatively large T difference (Page 13 L22) before or after the adjustment? If before, what is the temperature offset after the adjustment? Please also present some standard deviation of the hourly/daily/monthly T values using the mean monthly gradients. What is about thermo-dynamics considering relative humidity and saturation for calculating lapse-rates. Please give some more detailed information in section 3.2.1. and the results.

(C) Detailed Comments (P = page, L = line)

P1 L23: Please present the longitude in addition to the latitude.

P1 L23: Here an throughout the text: Above sea level can be abbreviated by "a.s.l."

P1 L23: Are the two years contrasting in hydrology or meteorology (or both).

P1 L27: Replace "increased by 100%" with "doubled"

P2 L3: Replace "cryosphere" with "glaciers(?) and the seasonal snow cover"

P2 L5: "winter months". Please consider to present these months in the introduction (June, July, August?!)

P2 L5: "intermittent": This can also mean at regular intervals, but I think you mean "erratic"?

P2 L24: Delete "evolution"

P3 L6: Revise to one relatively wet and one d

P3 L15: Remove "over time". Instead you can present a date to which the snow cover duration persist (which month/season?)

P3 L18: Add the longitude.

P3 L20: "best instrumented": Please explain in more detail. In contrast to which other catchments? Or just write "well-equipped" or "mounted"

P3 L21: Remove the "$\sim$" (also throughout the manuscript)

P3 L23: Remove "clean"

P3 L25: Please use 10x m$^3$ instead of Mm$^3$

P3 L26: Revise "rate" to "mean annual precipitation"

P3 L31: Is "area" the "study site"?

P3/ffP4: This sentence is hard to understand. Are these trajectories "storm paths". Please give some more detail. Also the sentence can be condensed to "The seasonal variability and frequency of precipitation events is also affected by precipitation trajectories..."

P5 L7: Remove "Finally"

P5 L5: Replace "for a specific campaign" by "next to the glacier"

P8 L12: Shift "the" to the front of La Laguna

P8 L13: Shift the reference to Figure one to the sentence before "the La Laguna catchment (Figure 1)"

P9 L9ff: Please consider to remove the subsections and to highlight the sub-model description by paragraphs and the sub-model names by italic font.

P10 L 8: Please unify "T" or "Tair"

P10 L10: Replace "please" by "we"

P10 L28: Add "snow albedo" to "minimum"

P12 L3: Add "sublimation" in front of "rate"

P12 L 19: Since the abbreviations have been introduced, use SWi here. Please check this throughout the text.

P12 L 19: You present absolute values here, but how much is this in % of mean SWi?

P12 L 20: Why is it in agreement? Why are weather conditions with more clouds necessarily colder? Whats about clear sky conditions at night causing very low temperatures?

P13 L1: Is this comparison performed before or after the Barne-downscaling? Please clarify here.

P13 L25: Correct "annual"

P14 L8: Revise this sentence to: "Simulated snow depths using... agreement with

measured snow depth values.

P15 L1: "forcings indicates": Remove one "s"

P19 L9: Please give more detail ion the time period and spatial extent of the averaged values here

P21 L7: Delete "when the snow...."

P25 L5: Revise to: Precipitation is known to be over-estimated using the WRF model

P25 L6: Correct "exist"

P25 L10: Please revise this sentence. Suggestion: Precipitation measurements using rain gauges can be biased towards an underestimation because of an undercatch particularly of snowfall due to wind influence.

P26 L26: Here the sublimation rate (absolute values) should be compared.

P27 L3: There will be no sublimation without a snow cover. Thus, this sentence is redundant. Perhaps you want to say that the snow cover duration SCD has a significant influence on sublimation/melt ratio.

P28 L17: It is rather the mass and energy balance of the snow cover, which includes sublimation.

P29 L19: Precipitation data is not an uncertainty, but the uncertainty of measured/modelled precipitation is.

Figure 1:

- Please include the reservoir and the glacier in the legend. Please adjust the elevation in the legend to the colour transitions.

Figure 3:

- Please unify the units. I would suggest Wm-2 and ms-1

- Please present in addition the snow depth, since this is an important parameter of this study

- Caption: Since only two years are presented, no "climatic" conditions are shown. Please revise to "meteorological conditions at the..."

- Please uniform the radiation abbreviations throughout the figures/manuscript to avoid confusion between incoming (index i) outgoing etc.

- Remove the "s" from "precipitations"

Figure 4

- Caption: Replace 'studied' with "La Laguna"

- Please describe in the caption which output is subtracted from the other for interpretation of the sign on the differences?

Figure 5

- I would suggest to bring the graphs of AWS/WRF-forced SD and observations in one figure for each station, and thus reduce the number of subfigures to 6.

Figure 6

- Please bring the decimal order (10ˆ8) to the label of the y-axes. I suggest to use km$^2$ like in the RMSE

Figure 7

- Modelled "energy" fluxes are shown

---

## Referee Comment (RC2) · Jonathan Conway (Referee) · 6 May 2019

Review of manuscript "Uncertainties in the spatial distribution of snow sublimation in the semi-arid Andes of Chile" by Marion Réveillet, Shelley MacDonell, Simon Gascoin, Christophe Kinnard, Stef Lhermitte and Nicole Schaffer.

Jono Conway

**General Comments:**

This paper presents an assessment of sublimation and melt rates from the snowpack in a semi-arid region of Chile. A distributed physically-based snow model is used to simulate snowpack evolution and quantify losses from sublimation and melt. The model is forced with two datasets that are spatialized on a 100m grid – one created from *in-situ* meteorological observations, the other from meteorological model output. The simulations are validated against in-situ snow depth measurements as well as snow cover observed using the MODIS platform. The results highlight the complex interactions between meteorological forcing and surface energy and mass balances, along with the effects of spatio-temporal averaging when calculating catchment-wide metrics in areas with predominately ephemeral snow cover. The rationale is well developed, the methods are fit-for-purpose and generally well explained, and many interesting and relevant results are presented.

However, there are too many lines of inquiry that compete for the readers attention and the figures presented do not always clearly present the main conclusions reached. In places, the results seem to contradict each other, and it not always clear in what direction different factors affect the sublimation ratio. For example, increased precipitation is attributed to increased melt based on Figure 12, but to increased sublimation based on the WRF simulations). This all contributes to make the paper quite hard to follow and reduces the confidence in the conclusions reached. The aims and hypothesis need to be clarified and perhaps re-assessed given the results available to ensure a coherent story can be told from the results. I would suggest the authors have a few options to refine and narrow the aim of the paper:

- If the aim is to understand the spatio-temporal variability of sublimation and melt, then only the simulations with AWS forcing should be presented as they appear to show the best validation, and the inclusion of the WRF result only confuse the reader. The results relating to season and effect of SCD and elevation can be highlighted.
- If the aim is to compare the effect of using the different forcing datasets (which is a valid aim given that methods to extrapolate beyond areas with in-situ measurements are needed), then the paper should be framed in this way and some of the later results should either be amended to include WRF results or removed (i.e. fig 11 & 12).
- If the aim is to establish the uncertainty in modelling actual sublimation and melt rates, then a more systematic approach requiring further simulations would be needed.

Of course, the authors may have other ways they wish to reframe, these are just some suggestions.

Additionally, some important aspects of the simulation deserve more attention:

- Validation needs to include comparison of wind speed and incoming radiative fluxes, as well as modelled surface temperature and albedo where possible.

- Greater discussion of wind speed and surface temperature as a key control on sublimation in the main body of the paper – e.g. the sensitivity of sublimation ratio to 10% change in wind speed noted in S4-15.
- Clearer description of how average results are calculated, especially the sublimation ratio vs ablation rates, considering the changing temporal and spatial scales associated with the ephemeral snow cover, and their effect on the results. The authors may wish to consider reporting surface energy balance terms as sums in MJ rather than rates (m yr$^{-1}$ or W m$^{-2}$) as these can be ambiguous when the you do not average over the full period (e.g. only over snow covered surfaces).
- Elevational gradients of SEB components need to be shown as it currently very hard to sort out different mechanisms for change in sublimation ratio (e.g. change in meteorology, snow covered area, elevational gradient). Ideally these would come before the catchment-average results to set the context for the observed inter-annual and/or inter-model differences.

With a more focussed aim and the clarification of these points, I have no doubt that this paper will make a good contribution to the literature.

**Specific Comments (page-line)**

3-21 The maximum elevation of the catchment is listed as 5630 m, but figures 4,7,10 and 11 show elevation bins > 6000m. Please clarify.

7-1 The figure caption describes these as hourly average values, but from the look of things (especially the SW) these appear to be daily averages. Please check and revise text.

8-13 Please indicate what height were the WRF output were output and if any scaling was used to transform their heights in Micromet.

10-30 Please indicate what sites and periods were used to choose the snow albedo values.

11-26 Please provide a fuller description of the kappa statistic as it is not a commonly used metric.

12-5 The definition of the sublimation ratio is not clear – is it the ratio of ablation totals, or sublimation vs melt rates (which depend on whether the surface is snow covered or not). This ambiguity becomes apparent later (see note 23-1). Please clarify in the text, perhaps with an equation.

13-1 The comparison of WRF simulations with AWS measurements needs to be shown if they are to be discussed. A table showing validation metrics (mean bias, mean absolute error etc) for different variables at each site used would be useful.

13-21 Please include comparison of WS, SWin, LWin in this section and Figure 4. These inputs are critical to the simulation of sublimation through the latent heat flux, surface temperature, and albedo.

13-22 Figure 4 would be more insightful if the actual mean values for T, RH etc were plotted for each forcing, rather than just the differences. E.g. Is the difference in precip because AWS precip decreases with height or WRF precip increases with height?

14-6 Model validation. Were any LWout or Ts measurements available for validation? A comparison of modelled vs measured surface temperature would strengthen the validation of turbulent flux and

subsurface scheme choices, which is a key area of uncertainty (as shown later by the sensitivity to z0).

16-1 In the caption, please indicate that periods of the validation data are missing. The green colour is hard to distinguish from the black, thus it appears the AWS simulation performed poorly at the three Tapado sites in late 2015 when there is a data gap. Consider using a different colour for the observed snow depth.

18-6. Please state what threshold was used to designate a snow-covered grid point in the model (e.g. 0.005 m w.e.). This can have a large bearing on the snow cover duration results, especially for small snowfalls such as those produced by WRF.

18-6 Please explain how data were averaged spatial and temporally (e.g. the average snow cover duration calculated for each individual grid point in the elevation band? Or the average of the grid cells that correspond with the modis pixels in each elevation band).

18-7 "Better performances were obtained for the AWS-forcings" while this is strictly correct, I don't think this comment is balanced. In 2015 the simulations are comparable and the improvement with the AWS forcing is minor.

19-9 Because elevation seems to have a greater effect on the sublimation rate, it would be useful to present these results (figure 10) before presenting the SEB and sublimation ratio results that are calculated over the whole catchment for snow-covered points only (Fig 8, 9). This would give better context for the somewhat complex interactions between SCA, SCD and meteorology. It would also be very useful to show the SEB results averaged in elevation bins after figure 10 to show reasons why the WRF simulations have higher sublimation.

19-11 Do the annual means include periods with no snow as 0 values? Please clarify how the annual means are calculated?

19-12 "For the mean AWS-simulation net SW is 24 and 23" do you mean the WRF-simulation here?

19-14 Do you mean -6 and -7 $Wm^{-2}$? These figures represent fairly small losses compared to mid-latitude sites (e.g. -25 to -20 $Wm^{-2}$ in Giesen et al, 2009), which is presumably due to the cold surface temperature of the snow surfaces.

22-1 Are these figures monthly average amounts for only snow-covered grid cells or something else? Please explain the averaging procedure in the methods section.

23-1 Figure10 – the sublimation ratio and ablation rates do not seem to match up – the ratio at high elevations is ~100% which implies there is little melt, but for both AWS and WRF forcing there is still a significant melt rate, and for AWS in 2015, melt rate==sublimation rate. Is this an artefact of the averaging of melt rate over snow only? Also, why do melt rates increase with height? Perhaps you are better to present the ablation totals rather than the ablation rates? Either way, the top and bottom panels should be consistent.

26-30 It is not clear what you mean by evaposublimation? Do you mean the evaporation of liquid from a melting surface or a combination of this process + sublimation from a frozen surface? If you are including evaporation from a melting surface, then doesn't the increased rate occur because the melting lowers the latent heat required to transform the water to vapour? Please be more explicit about the process occurring.

Figure S2 If this precipitation sensitivity analysis is retained, Figure S2 needs to be included in the results section as it is discussed directly.

**Additional references**

Giesen RH, Andreassen LM, Van den Broeke MR and Oerlemans J (2009) Comparison of the meteorology and surface energy balance at Storbreen and Midtdalsbreen, two glaciers in southern Norway. Cryosphere, 3(1), 57–74, doi: 10.5194/tc-3-57-2009

---

## Author Comment (AC1) · 10 Jun 2019

**Response to Reviewer #1**

**A) General Comments**

*In the manuscript "Uncertainties in the spatial distribution of snow sublimation in the semi-arid Andes of Chile" Réveillet et al. present results of their study aiming to simulate melt and sublimation rates over the instrumented watershed of La Laguna. They present the relative importance of sublimation versus snow melt using a distributed snowpack model for two meteorologically contrasting years. They detected a large difference in modelled sublimation rates forcing the model with data of AutomaticWeather Stations (AWS) and with Weather Research and Forecasting (WRF) model data. This difference is caused by (a) the different meteorological input, particularly in precipitation and temperature, and (b) by the modelled snow cover persistence. The objective of the study of Réveillet et al. is to assess the uncertainties in melt and sublimation arising from modelling snow evolution using AWS and/or WRF-model generated meteorological datasets. Since the seasonal snow cover and glacier melt as processes of the cryosphere have locally a high contribution to available fresh water in this region. The study of Réveillet et al. contributes to gain knowledge of snow depth distribution and snow cover processes in the semi-arid Andes of Chile, and thus the manuscript is in the scope of TC. In general, this study shows again the importance of a critical interpretation of model results with respect to model input data. The introduction is complete, the applied methods are appropriate and comprehensible, and the results are compared to referenced work. However, the manuscript at its present stage summarizes results of interesting work packages and analysis, but lacks in the overarching aim with a clear problem statement, the research questions and the respective conclusions. This is also obvious in the high number of subsections presenting methods and results which not necessarily contribute to the conclusions. My suggestion is to restructure the manuscript with a clear focus on formulated research questions and to revise the title reconsidering the term "uncertainty".*

**Authors'answer:**

We thank the reviewer for this constructive and thorough review of the manuscript.
As requested, and in agreement with comments made by the reviewer #2, the manuscript has been restructured and now focuses on the differences in simulated sublimation depending on the forcing used. More information is provided in the detailed response to Specific comment #1.

**B) Specific Comments**

*(1) As addressed in the general comments, the manuscript shows an interesting work but without presenting an overarching aim and respective research questions. Forcing the model with different data from AWS and WRF does not result only in differences of sublimation rates, but also differences in e.g. snow covered area, snow persistence and snow melt. So I would suggest that not the uncertainties in the spatial distribution of snow sublimation are shown. Rather the variation of snow related parameters and processes forcing the applied model with different input is presented. Since the weaknesses of the WRF model output are known (cold bias, precipitation overestimation) and the AWS data might be the more appropriate model input, the study shows the error of forcing the model with WRF data for snow parameters (i.e. the overestimation of the snow cover duration by 2 month). The impacts of these errors are particularly obvious in the sublimation rates and ratios, which again are a function of snow coverage, elevation, temperature, etc. If the (model?!) uncertainty in sublimation should be addressed in more detail, I would suggest to show at least one additional figure presenting the calculated and simulated sublimation rates at AWS locations. In my opinion, the manuscript at its*

*present stage presents "Differences in simulated sublimation in a high mountain catchment of the semi-arid Andes of Chile using AWS data and WRF meteorological forcing". Possible research question may address the main differences (errors?) in sublimation simulated using the WRF forcing, the impact of SCA and SCD under/overestimation, and the effect of different meteorological conditions of the two contrasting years on sublimation ratios. This can lead to conclusions about advantages and disadvantages of using the different forcing data. Most of the results are already presented, but the paper should be restructured accordingly to answer the formulated research questions and further drawing the conclusions.*

**Authors'answer:** To address the comments of both reviewers, the paper has been restructured (including changing the title) to focus more on the differences in terms of snow depth, snow cover and sublimation as a result of the chosen forcing. This modification includes a change in result section, where a better comparison between the two forcings is provided (i.e. section 4.2 renamed to "Snow depth and snow cover comparison" for consistency).
The results sections of the original manuscript have also been revised and organized in:
4.3 Ablation and energy balance fluxes
4.3.1 Mean annual elevation gradients
This includes changes to Figure 10 and the addition of a new figure (Fig. 9) that show the energy fluxes contribution with respect to elevation (as suggested by reviewer #2)
4.3.2 Monthly evolutions
In this section, Figures 9 and 8 are presented and commented Figures 9 and 8.
The discussion section has also been re-organized with a stronger focus on (i) differences in sublimation as a function of the chosen forcing (ii) differences in sublimation between the two years and (iii) the impact of snow depth on the sublimation ratio and (iii) the limits of the study.
This re-organisation was chosen to focus the conclusions on advantages and disadvantages of using the different forcing data. For that purpose, differences in terms of meteorological data and the consequences on SD, SC and sublimation are studied, as suggested in comment #1.

*(2) Two years with contrasting meteorological conditions have been chosen for this study. This has the advantage of testing the simulation results, but also seems to be restricted to AWS data availability. To get an overview of the overall climate in this region and to classify the two selected years in a climatological context, please present a short climate overview of the last 30 years from a nearby station, or at least some short statistics for the AWS with the longest data history (La Laguna?)*
**Authors'answer:** To address comment #2, a figure showing the monthly mean air temperature and precipitation over the 1976-2016 period recorded at la Laguna has been added to the supplementary information. The 2014 and 2015 measurements have been included in this figure.

[Figure]

*Figure S1: Monthly precipitation (a) and air temperature (b) recorded at La Laguna station. The monthly mean (black) is computed over the 1976-2016 period.*

**(3)** *The downscaling of the WRF data still appears opaque to me. Is the relatively large T difference (Page 13 L22) before or after the adjustment? If before, what is the temperature offset after the adjustment? Please also present some standard deviation of the hourly/daily/monthly T values using the mean monthly gradients. What is about thermo-dynamics considering relative humidity and saturation for calculating lapse-rates. Please give some more detailed information in section 3.2.1. and the results.*

**Authors'answer:** We agree that the section 4.1.2 dealing with the comparison between the forcing was confusing. As the paper has been re-structured and now focuses in more details on the impact of the forcing choice on the simulation, the forcing data comparison is important. Modifications have been made in this section accordingly.

1- We agree that having the comparison before running the *Micromet* subroutine is interesting. Nevertheless, the direct comparison remains complicated in this study, mainly due to the spatial offset (and especially the vertical difference) between the AWS location and the closest WRF grid point. This is why we chose to present in the paper the comparison at the catchment scale from the *MicroMet* outputs to overcome this issue. As this information remains important, it is now mentioned in the manuscript and the vertical offset is available in the Table S1 in the supplementary material. In addition, a Table showing validation metrics (R2, RMSE and Absolute mean error) between the AWS measurements and the outputs from the closest WRF grid-point after running *MicroMet* is provided in the supplementary material.

 "Details and statistics information about the comparison at each AWS locations are available in Table S1 (in the supplementary material). Note that here the comparison between the AWS measurements and the closest WRF grid point is not presented due to the significant vertical offset between the two points (Table S1 in the supplementary material)."

2- We don't really understand your request. Could you please clarify this point? Are you talking about the standard deviation for each station for each moth between the AWS measurements and the *Micromet* outputs forced by the AWS-forcing? Or a cross-validation study?

In addition, measurements have been used to compute the laspe rates, but as results were close to the default values, it has been chosen to keep the default parameterization to run the model. These information are given in section 3.2.1 :

"Spatial interpolation using the Barnes scheme was used to distribute the nine AWS measurements of T, RH, LWi, SWi and pressure over the model domain. As relative humidity is a non-linear function of elevation, the relatively linear dewpoint temperature is used for the elevation adjustment. For more details refer to Liston and Elder (2006). In this study the *MicroMet* subroutine has been run with the default setting for the Southern Hemisphere, for air temperature and dewpoint temperature monthly lapse rates (Liston and Elder, 2006b). Monthly lapse rates computed from the available measurements are dependent on the year considered. As the mean is close to the default settings, it has been chosen to conserve these values."

3- Section 3.2.1 has been clarified as follow:

"The 3km WRF outputs (i.e. the 240 points (Figure 1) decribed in section 2.2.1) were used as inputs for *MicroMet* which considers that each WRF cell corresponds to a virtual weather station located in the center of the WRF cell, following Mernild et al. (2017) and Baba et al. (2018a). In other words, this means that MicroMet has been forced by 240 virtual stations containing the WRF outputs meteorological dara. As an vertical offset exists between the WRF grid point elevation and the DEM, *MicroMet* adjusts this offset at the corresponding coordinate and downscales the data to a 100 m grid."

The result part was also modified as mention above.

**C) Detailed Comments**

*P1 L23: Please present the longitude in addition to the latitude.*
**Authors'answer:** This information has been added: 70°W

*P1 L23: Here an throughout the text: Above sea level can be abbreviated by "a.s.l."*
**Authors'answer:** Done

*P1 L23: Are the two years contrasting in hydrology or meteorology (or both).*
**Authors'answer:** This is a difficult question given that long-term gauges are located downstream of a dam (La Laguna). It's likely both, but in this study we are more focused on precipitation amounts. We have specified this in the manuscript:
**"…**La Laguna (3150–5630 m a.s.l., 30°S 70°W), during two hydrologically contrasting years (i.e. dry *vs.* wet)."
In addition, in the introduction and section 2 a reference to the figure S1 has been added.

*P1 L27: Replace "increased by 100%" with "doubled"*
**Authors'answer:** Done

*P2 L3: Replace "cryosphere" with "glaciers(?) and the seasonal snow cover"*
**Authors'answer:** Done

*P2 L5: "winter months". Please consider to present these months in the introduction (June, July, August?!)*
**Authors'answer:** It has been defined as follows: **"**that are largely limited to winter months (i.e. June, July and August)"

***P2 L5:*** *"intermittent": This can also mean at regular intervals, but I think you mean "erratic"?*
**Authors'answer:** Yes, "intermittent" has been changed to "erratic"

***P2 L24:*** *Delete "evolution"*
**Authors'answer:** Done

***P3 L6:*** *Revise to one relatively wet and one d*
**Authors'answer:** The sentence was modified accordingly.

***P3 L15:*** *Remove "over time". Instead you can present a date to which the snow cover duration persist (which month/season?)*
**Authors'answer:** "over time" has been changed to "at the end of the winter season (i.e. in August, September)", to address this comment.

***P3 L18:*** *Add the longitude.*
**Authors'answer:** The longitude has been specified as in the abstract.

***P3 L20:*** *"best instrumented": Please explain in more detail. In contrast to which other catchments? Or just write "well-equipped" or "mounted"*
**Authors'answer:** To address this comment, we have specified as follows: "…is the most instrumented within the region"

***P3 L21:*** *Remove the "~" (also throughout the manuscript)*
**Authors'answer:** "~" has been removed throughout the manuscript.

***P3 L23:*** *Remove "clean"*
**Authors'answer:** Done

***P3 L25:*** *Please use 10x m3 instead of Mm3*
**Authors'answer:** Done, in addition the value was wrong. Therefore, "200 Mm$^3$" has been change by "$38.10^6$ m$^3$".

***P3 L26:*** *Revise "rate" to "mean annual precipitation"*
**Authors'answer:** Done

***P3 L31:*** *Is "area" the "study site"?*
**Authors'answer:** Yes, this has been specified.

***P3/ffP4:*** *This sentence is hard to understand. Are these trajectories "storm paths". Please give some more detail. Also the sentence can be condensed to "The seasonal variability and frequency of precipitation events is also affected by precipitation trajectories..."*
**Authors'answer:** The sentence has been re-written: "Seasonal precipitation variability and frequency are also complicated by individual storm trajectories (e.g. Sinclair and MacDonell, 2016) which can cause large differences in relative precipitation distribution across the catchment, a phenomenon also described in central Chile (Burger et al., 2019)."
And the following reference has been added:
Burger, F., Ayala, A., Farias, D., Shaw, T. E., MacDonell, S., Brock, B., McPhee, J. and Pellicciotti, F.: Interannual variability in glacier contribution to runoff from a high elevation Andean catchment:

understanding the role of debris cover in glacier hydrology. *Hydrological Processes*, 33(2), 214-229. https://doi.org/10.1002/hyp.13354, 2019

*P5 L7: Remove "Finally"*
**Authors'answer:** Done

*P5 L5: Replace "for a specific campaign" by "next to the glacier"*
**Authors'answer:** According to your comment, "for a specific campaign" has been replaced by "in the debris-covered part of the glacier"

*P8 L12: Shift "the" to the front of La Laguna*
**Authors'answer:** Done

*P8 L13: Shift the reference to Figure one to the sentence before "the La Laguna catchment (Figure 1)"*
**Authors'answer:** Done

*P9 L9ff: Please consider to remove the subsections and to highlight the sub-model description by paragraphs and the sub-model names by italic font.*
**Authors'answer:** Done

*P10 L8: Please unify "T" or "Tair"*
**Authors'answer:** "T" has been chosen and is now used over the entire manuscript.

*P10 L10: Replace "please" by "we"*
**Authors'answer:** Done

*P10 L28: Add "snow albedo" to "minimum"*
**Authors'answer:** Done

*P12 L3: Add "sublimation" in front of "rate"*
**Authors'answer:** Done

*P12 L19: Since the abbreviations have been introduced, use SWi here. Please check this throughout the text.*
**Authors'answer:** "SWi" is now used in the manuscript (and in the Figures) after being defined, as well as "LWi".

*P12 L19: You present absolute values here, but how much is this in % of mean SWi?*
**Authors'answer:** This information have been added for SWi and LWi. The paragraph has been re-written as follows:
"According to the AWS measurements, Jan-Jul 2015 was warmer than Jan-Jul 2014. Conversely, observations indicate lower temperatures for Aug-Dec 2015 than for Aug-Dec 2014 (daily mean difference of -2.6°C). Relative humidity was higher for 2015 compared to 2014 (daily mean difference of 11%) whereas SWi was lower (mean difference of -18 W m$^{-2}$, i.e. 6% of the mean SWi), with larger differences in Jul-Dec (daily mean difference of -32 W m$^{-2}$, i.e. 12% of the daily mean SWi), and LWi was higher (daily mean difference of 20 W m$^{-2}$, i.e. 7% of the daily mean LWi). This decrease SWi and increase in LWi can be explained by a larger number of clouds in 2015."

***P12 L20:*** *Why is it in agreement? Why are weather conditions with more clouds necessarily colder? Whats about clear sky conditions at night causing very low temperatures?*
**Authors'answer:** We agree that the relationship between SW and air temperature is more complex. To avoid confusion and considering that this information is not central to the paper, the sentence has been deleted.

***P13 L1:*** *Is this comparison performed before or after the Barne-downscaling? Please clarify here.*
**Authors'answer:** Yes, it was performed before the Barne-downscaling. Nevertheless this section is confusing. In agreement with your specific comment 3 and remarks made by the second reviewer, this information is now in the supplementary information as a Table."

***P13 L25:*** *Correct "annual"*
**Authors'answer:** Done

***P14 L8:*** *Revise this sentence to: "Simulated snow depths using. . . agreement with measured snow depth values.*
**Authors'answer:** The sentence has been changed according to your comment. It now reads: "Simulated snow depths using the AWS-forcings (Figures 5 a-f) are in good agreement with measured snow depth values (mean $k$=0.14 and mean $RMSE$=0.15 m)."

***P15 L1:*** *"forcings indicates": Remove one "s"*
**Authors'answer:** Done.

***P19 L9:*** Please give more detail ion the time period and spatial extent of the averaged values here
**Authors'answer:** This sentence has been restructured and the issue in question is no longer relevant here. However, in order to address this comment and remarks made by reviewer #2, more details have been added in the methods section as follow: "Note that sublimation and energy balance are only computed over snow surfaces. This means that annual and monthly means are only computed at grid-cells with snow."

***P21 L7:*** *Delete "when the snow. . .."*
**Authors'answer:** Done

***P25 L5:*** *Revise to: Precipitation is known to be over-estimated using the WRF model*
**Authors'answer:** Done

***P25 L6:*** *Correct "exist"*
**Authors'answer:** Done

***P25 L10:*** *Please revise this sentence. Suggestion: Precipitation measurements using rain gauges can be biased towards an underestimation because of an undercatch particularly of snowfall due to wind influence.*
**Authors'answer:** *Done*

***P26 L26:*** *Here the sublimation rate (absolute values) should be compared.*
**Authors'answer:** We agree with your comment and also think that having these rates will also be very useful for the discussion. Therefore, this information has been added to the results section as

follows:

"The mean daily rate is 0.6 mm w.e. d$^{-1}$ and 3.6 w.e. d$^{-1}$ for 2014 and 2015, respectively, when the model is forced with the AWS-forcing. Values are larger and reach 3.1 mm w.e. d$^{-1}$ and 4.1 mm w.e. d$^{-1}$ for 2014 and 2015 when simulations are performed with the WRF-forcing."

The discussion has been entirely re-organized and all the sections in 5.2 have been re-written. Sublimation rates are now discussed in this section.

***P27 L3:*** *There will be no sublimation without a snow cover. Thus, this sentence is redundant. Perhaps you want to say that the snow cover duration SCD has a significant influence on sublimation/melt ratio.*
**Authors'answer:** This sentence has been removed due to the restructuration and to avoid confusion.

***P28 L17:*** *It is rather the mass and energy balance of the snow cover, which includes sublimation.*
**Authors'answer:** This sentence has been modified accordingly.

***P29 L19:*** *Precipitation data is not an uncertainty, but the uncertainty of measured/modelled precipitation is.*
**Authors'answer:** The sentence has been re-written to address this comment: "First, the main forcing uncertainty is from precipitation due to measurement errors and lack of spatial representation as precipitation data was only available for two stations."

**Figure 1:**
- Please include the reservoir and the glacier in the legend. Please adjust the elevation in the legend to the colour transitions.
**Authors'answer:** Done

**Figure 3:**
- Please unify the units. I would suggest Wm-2 and ms-1
- Please present in addition the snow depth, since this is an important parameter of this study
- Caption: Since only two years are presented, no "climatic" conditions are shown.
Please revise to "meteorological conditions at the. . ."
- Please uniform the radiation abbreviations throughout the figures/manuscript to avoid confusion between incoming (index i) outgoing etc.
- Remove the "s" from "precipitations"
**Authors'answer:** Figure 3 and the caption have been revised to address this comment.

**Figure 4:**
- Caption: Replace 'studied' with "La Laguna"
- Please describe in the caption which output is subtracted from the other for interpretation of the sign on the differences?
**Authors'answer:** The caption has been changed. The Figure has been modified according to the reviewer #2 comment and the variables are now plotted instead of the differences.

[Figure]

*Figure 4: (a) Area-elevation distribution of the La Laguna catchment. (b to g) MicroMet outputs at the catchment scale forced by the AWS (red) and the WRF (blue) for 2014 (lines) and 2015 (dashed lines).*

**Figure 5**

- I would suggest to bring the graphs of AWS/WRF-forced SD and observations in one figure for each station, and thus reduce the number of subfigures to 6.

**Authors'answer:** We initially did this, but it was difficult to distinguish AWS-forced SD and WRF-forced SD and almost impossible to compare it to the observation. So we decided to split it into different graphs.

**Figure 6**

- Please bring the decimal order (10ˆ8) to the label of the y-axes. I suggest to use km2 like in the RMSE

**Authors'answer:** Done, km2 is now used.

**Figure 7**

- Modelled "energy" fluxes are shown

**Authors'answer:** Done

---

## Author Comment (AC2) · 10 Jun 2019

**Response to Reviewer #2**

**A) General Comments**

This paper presents an assessment of sublimation and melt rates from the snowpack in a semi-arid region of Chile. A distributed physically-based snow model is used to simulate snowpack evolution and quantify losses from sublimation and melt. The model is forced with two datasets that are spatialized on a 100m grid – one created from in-situ meteorological observations, the other from meteorological model output. The simulations are validated against in-situ snow depth measurements as well as snow cover observed using the MODIS platform. The results highlight the complex interactions between meteorological forcing and surface energy and mass balances, along with the effects of spatio-temporal averaging when calculating catchment-wide metrics in areas with predominately ephemeral snow cover. The rationale is well developed, the methods are fit-for-purpose and generally well explained, and many interesting and relevant results are presented.

**1)** However, there are too many lines of inquiry that compete for the readers attention and the figures presented do not always clearly present the main conclusions reached. In places, the results seem to contradict each other, and it not always clear in what direction different factors affect the sublimation ratio. For example, increased precipitation is attributed to increased melt based on Figure 12, but to increased sublimation based on the WRF simulations). This all contributes to make the paper quite hard to follow and reduces the confidence in the conclusions reached. The aims and hypothesis need to be clarified and perhaps re-assessed given the results available to ensure a coherent story can be told from the results. I would suggest the authors have a few options to refine and narrow the aim of the paper:
- If the aim is to understand the spatio-temporal variability of sublimation and melt, then only the simulations with AWS forcing should be presented as they appear to show the best validation, and the inclusion of the WRF result only confuse the reader. The results relating to season and effect of SCD and elevation can be highlighted.
- If the aim is to compare the effect of using the different forcing datasets (which is a valid aim given that methods to extrapolate beyond areas with in-situ measurements are needed), then the paper should be framed in this way and some of the later results should either be amended to include WRF results or removed (i.e. fig 11 & 12).
- If the aim is to establish the uncertainty in modelling actual sublimation and melt rates, then a more systematic approach requiring further simulations would be needed.
Of course, the authors may have other ways they wish to reframe, these are just some suggestions.

Additionally, some important aspects of the simulation deserve more attention:
**2)** Validation needs to include comparison of wind speed and incoming radiative fluxes, as well as modelled surface temperature and albedo where possible.
**3)** Greater discussion of wind speed and surface temperature as a key control on sublimation in the main body of the paper – e.g. the sensitivity of sublimation ratio to 10% change in wind speed noted in S4-15.
**4)** Clearer description of how average results are calculated, especially the sublimation ratio vs ablation rates, considering the changing temporal and spatial scales associated with the ephemeral snow cover, and their effect on the results. The authors may wish to consider reporting surface energy balance terms as sums in MJ rather than rates (m yr-1 or W m-2) as these can be ambiguous when the you do not average over the full period (e.g. only over snow covered surfaces).

**5)** Elevational gradients of SEB components need to be shown as it currently very hard to sort out different mechanisms for change in sublimation ratio (e.g. change in meteorology, snow covered area, elevational gradient). Ideally these would come before the catchment-average results to set the context for the observed inter-annual and/or inter-model differences.

With a more focussed aim and the clarification of these points, I have no doubt that this paper will make a good contribution to the literature.

**Authors'answer:**
Thank you for your positive comments and interest in the paper. We have carefully answered each point presented in the general comments as well as the specific/technical points. Changes made in response to reviewer comments in the manuscript are in green.

**1) Authors'answer:** To address the comments of both reviewers, the paper has been restructured (including changing the title) to focus more on the differences in terms of snow depth, snow cover and sublimation as a result of the chosen forcing. This modification includes a change in result section, where a better comparison between the two forcings is provided (i.e. section 4.2 has been renamed to "Snow depth and snow cover comparison**"** for consistency).
The results sections of the original manuscript have also been revised and organized as follows:
4.3 Ablation and energy balance fluxes
4.3.1 Mean annual elevation gradients
This includes changes to Figure 10 and the addition of a new figure (Fig. 9) that show the energy fluxes contribution as a function of elevation (as suggested by reviewer #2)
4.3.2 Monthly evolutions
In this section, Figures 9 and 8 are presented and commented Figures 9 and 8.
The discussion section has also been re-organized with a stronger focus on (i) differences in sublimation as a function of the chosen forcing (ii) differences in sublimation between the two years and (iii) the impact of snow depth on the sublimation ratio and (iii) the limits of the study.
This reorganization was chosen to focus the conclusions on advantages and disadvantages of using different forcing data. For that purpose, differences in terms of meteorological data and the consequences on SD, SC and sublimation are analyzed, as suggested in your comment.

**2)** The comparison of wind speed and radiative fluxes has been added in section 4.1.2 and Figure 4 (in agreement with the specific comments P13 L1, L21 and L22). For more details please refer to these comments.
Regarding the albedo validation, measurement data have been used to calibrate the model. Information about the available measurements is given by Figure 2. In addition, this point has been clarified according to your comment P10 L30)
Regarding the surface temperature, we have chosen not to add a new graph. For more details please refer to comments P14 L6.

**3)** Due to the new reorganization of the paper, and a focus on the impact of different forcings on sublimation, we worry that this point will be a little bit disconnected from the main scientific question. Nevertheless since this point is important and interesting, we created a new section in the discussion, named 'Limits of the study', and the results are included in the supplementary material.

**4)** The calculation is done considering only days with snow on the ground. In order to address your comment and a comment made by the other reviewer, this information is now provided in the methods

section: "Note that sublimation and energy balance are only computed over snow surfaces. This means that annual and monthly means are computed at grid-cells with snow only."

In addition, we tried reporting surface energy balance terms as sums in MJ (see Figures below). However, we think that in this case the number of days changes the MJ sum and the graph therefore illustrates the number of days more clearly than the energy flux differences. For instance, for the elevation plot when fluxes at a given elevation are compared, larger values are found at higher elevations and this is related to the number of days rather than changes in the energy flux. Regarding the monthly average, larger values using the unit MJ are found for the months when snow was on the ground for all days.

The alternative would be to compute the MJ over a similar time period, but given the data availability for this catchment, this would reduce the results to a very short time period. Therefore, even though the average cannot be calculated over the entire time period, rather only when there is snow covering the surface, we think that using W m$^{-2}$ allows for a more accurate interpretation of the results.

[Figure]

Figure: Annual sum of main modeled energy fluxes (computed over snow surfaces only) for each 200 m elevation band using AWS (a,b) and WRF (c,d) forcing for 2014 (a,c) and 2015 (b,d). SW (green) is net shortwave radiation, LW (yellow) is net longwave radiation, QE (red) is the latent heat flux and QH (purple) is the sensible heat flux.

[Figure]

Figure: Monthly sum of the main modeled energy fluxes for the entire catchment, over snow surfaces only. SW (green) is net shortwave radiation, LW (yellow) is net longwave radiation, QE (red) is the latent heat flux and QH (purple) is the sensible heat flux

**5)** A Figure showing the energy fluxes against elevation has been added to the revised manuscript (Figure 9), before the catchment averages are discussed as suggested.

[Figure]

*Figure 9: Annual mean of main modeled energy fluxes (computed over snow surfaces only) for each 200 m elevation band using AWS (a,b) and WRF (c,d) forcing for 2014 (a,c) and 2015 (b,d). SW is net shortwave radiation, LW is net longwave radiation, QE is the latent heat flux and QH is the sensible heat flux.*

**B) Specific Comments**

***P3-L21*** *The maximum elevation of the catchment is listed as 5630 m, but figures 4,7,10 and 11 show elevation bins > 6000m. Please clarify.*
**Authors'answer:** Thank you for this observation, indeed, the maximum and the minimum elevation of the DEM presented in Figure 1 are respectively 6211 and 3143 m a.s.l.. This has been corrected in the revised manuscript.

***P7-L1*** *The figure caption describes these as hourly average values, but from the look of things (especially the SW) these appear to be daily averages. Please check and revise text.*
**Authors'answer:** Thank you for this observation; the caption is incorrect as daily data are plotted here. This has been corrected.

***P8-L13*** *Please indicate what height were the WRF output were output and if any scaling was used to transform their heights in Micromet.*
**Authors'answer:** The WRF outputs are 2 m above the surface. This information is an input into the *MicroMet* model, and the meteorological outputs are therefore at 2m. This information has been added in the manuscript as follows:
"The model outputs are at 2m above the surface and are available at 22 km resolution over Chile"

***P10-L30*** *Please indicate what sites and periods were used to choose the snow albedo values.*
**Authors'answer:** Albedo measurement availability is mentioned in Figure 2. All the measurements have been used to calibrate the model. This is point is now clarified clarified in the new manuscript:
"The minimum and maximum snow albedo (corresponding to old and fresh snow, respectively) are respectively fixed to 0.6 and 0.9 in agreements with all the measurements performed at the AWSs (Figure 2)."

***P11-L26*** *Please provide a fuller description of the kappa statistic as it is not a commonly used metric.*
**Authors'answer:** A more complete definition of kappa statistic has been added as follow:
"The performance was evaluated using a Kappa statistic coefficient (Cohen, 1960) denoted *k,* to measure the agreement between the simulation and the observation, considering the percentage of time with and without snow. The calculation of k is here performed according to the following formula:

$$k = \frac{\Pr(a) - \Pr(e)}{1 - \Pr(e)} \qquad (1)$$

where *Pr(a)* represents the actual observed agreement (i.e. snow or no snow for both simulation and observation); and *Pr(e)* represents the hypothetical probability of chance agreement. Complete agreement is defined when *k*=1."

***P12-L5*** *The definition of the sublimation ratio is not clear – is it the ratio of ablation totals, or sublimation vs melt rates (which depend on whether the surface is snow covered or not). This ambiguity becomes apparent later (see note 23-1). Please clarify in the text, perhaps with an equation.*
**Authors'answer:** This is now better defined in the method section:
"The sublimation ratio is defined as a percentage, and equal to the sublimation divided by the total ablation (i.e. sublimation and melt rates). Note that sublimation and energy balance are only computed over snow surfaces. This means that annual and monthly means are only computed at grid-cells with snow."

***P13-L1*** *The comparison of WRF simulations with AWS measurements needs to be shown if they are to*

*be discussed. A table showing validation metrics (mean bias, mean absolute error etc) for different variables at each site used would be useful.*

**Authors'answer:** We agree with this comment. Nevertheless, the direct comparison between the AWS measurements and the WRF outputs remains complicated in this study, mainly due to the spatial offset (and especially the vertical difference) between the AWS location and the closest WRF grid point. Therefore we chose to present the comparison at the catchment scale from the *MicroMet* outputs to overcome this issue. As this information remains important, it is now mentioned in the manuscript and the vertical offset is available in the Table S1 in the supplementary material.

In addition, a Table showing validation metrics (R2, RMSE and Absolute mean error) between the AWS measurements and the outputs from the closest WRF grid-point after running *MicroMet* is provided in the supplementary material.

"Details and statistics information about the comparison at each AWS locations are available in Table S1 (in the supplementary material). Note that here the comparison between the AWS measurements and the closest WRF grid point is not presented due to the significant vertical offset between the two points (Table S1 in the supplementary material)."

***P13-L21*** *Please include comparison of WS, SWin, LWin in this section and Figure 4. These inputs are critical to the simulation of sublimation through the latent heat flux, surface temperature, and albedo.*

**Authors'answer:** The figure has been modified (see below) and new panels (Figure 4 e,f,g ) have been added to the manuscript:

"The SWi and LWi remain very similar. The wind speed outputs differ (Figure 4e), especially above 4500 m a.s.l. where differences reach a maximum of 4 m s⁻¹. The comparison between the AWS measurements and the closest WRF grid point output yield similar results. However this comparison should be viewed with caution given that there is a spatial offset between the AWS location and the closest WRF grid (Table S1 in the supplementary material)."

[Figure]

Figure 4: (a) Area-elevation distribution of the La Laguna catchment. (b to g) MicroMet outputs at the catchment scale forced by the AWS (red) and the WRF (blue) for 2014 (lines) and 2015 (dashed lines).

***P13-L22*** *Figure 4 would be more insightful if the actual mean values for T, RH etc were plotted for each forcing, rather than just the differences. E.g. Is the difference in precip because AWS precip decreases with height or WRF precip increases with height?*

**Authors'answer:** Figure 4 now represents the actual mean values (see figure above).

*P14-L6 Model validation. Were any LWout or Ts measurements available for validation? A comparison of modelled vs measured surface temperature would strengthen the validation of turbulent flux and subsurface scheme choices, which is a key area of uncertainty (as shown later by the sensitivity to z0).*

**Authors'answer:** We completely agree with this remark. There are LWO measurements available at the Tapado AWS for the July to December 2015 period. The comparison of these measurements (when snow height > 0.05 m) to the MicroMet output shows that the model generally follows a similar pattern to the measured surface temperature, however consistently underestimates surface temperature (see below Figure). Measured and modeled surface temperatures are consistently below zero, which gives confidence in the use of the parameterization. We have chosen not to include this new Figure in the revised version as it would likely complicate the flow of the paper.

[Figure]

*P16-L1 In the caption, please indicate that periods of the validation data are missing. The green colour is hard to distinguish from the black, thus it appears the AWS simulation performed poorly at the three Tapado sites in late 2015 when there is a data gap. Consider using a different colour for the observed snow depth.*

**Authors'answer:** The color of the measurement has been changed to red. The areas shaded in green indicate the period with validation data so we are not sure that adding this information in the legend is necessary.

*P18-L6 Please state what threshold was used to designate a snow-covered grid point in the model (e.g. 0.005 m w.e.). This can have a large bearing on the snow cover duration results, especially for small snowfalls such as those produced by WRF.*

**Authors'answer:** The threshold used to designate a snow-covered grid point is fixed at 3mm w.e.. This information has been added in the method part as new section:

"The snow cover area (SCA) and the snow cover duration (SCD) over the entire catchment were compared to the MODIS product. A threshold of 0.003 m w.e. was used to convert the simulated SWE into snow presence or absence for each grid cell (within the same range as Gascoin et al., 2015). Since the MODIS SCA product corresponds to the maximum visible extent over a period of 8 days, we also computed the maximum SCA over the same 8 day period from the simulated SCA for comparison."

*P18-L6 Please explain how data were averaged spatial and temporally (e.g. the average snow cover*

*duration calculated for each individual grid point in the elevation band? Or the average of the grid cells that correspond with the modis pixels in each elevation band).*

**Authors'answer:** This has been clarified as follow:

"The simulated snow cover duration (SCD) was also compared to the observed duration (from MODIS) by elevation band. For all each 200 m elevation band, the total number of snow-covered days for each grid cell was computed and then averaged for each band."

***P18-L7*** *"Better performances were obtained for the AWS-forcings" while this is strictly correct, I don't think this comment is balanced. In 2015 the simulations are comparable and the improvement with the AWS forcing is minor.*

**Authors'answer:** We agree with this comment and the results are described for each year, allowing the reader to make this distinction: "For 2014, better performances were obtained for the AWS-forcing than for the WRF-forcing (Figure 7). For 2015, while better performances were also obtained for the AWS-forcing, the improvement using this forcing was minor."

***P19-L9*** *Because elevation seems to have a greater effect on the sublimation rate, it would be useful to present these results (figure 10) before presenting the SEB and sublimation ratio results that are calculated over the whole catchment for snow-covered points only (Fig 8, 9). This would give better context for the somewhat complex interactions between SCA, SCD and meteorology. It would also be very useful to show the SEB results averaged in elevation bins after figure 10 to show reasons why the WRF simulations have higher sublimation.*

**Authors'answer:** To address this comment, and in response to the general comment above, Figure 10 is now Figure 8 in the revised manuscript and comes before the SEB and sublimation ratio results computed for the entire catchment. In addition, the figure showing the energy contribution of each flux against elevation has been added:

[Figure]

*Figure 9: Annual mean of main modeled energy fluxes (computed over snow surfaces only) for each 200 m elevation band using AWS (a,b) and WRF (c,d) forcing for 2014 (a,c) and 2015 (b,d). SW is net shortwave radiation, LW is net longwave radiation, QE is the latent heat flux and QH is the sensible heat flux.*

***P19-L11*** *Do the annual means include periods with no snow as 0 values? Please clarify how the annual means are calculated?*

**Authors'answer:** The means are calculated only for period with snow. This information has been added in the methods section as follows:

"Note that sublimation and energy balance are only computed over snow surfaces. This means that annual and monthly means are only computed at grid-cells with snow."

***P19-L12*** *"For the mean AWS-simulation net SW is 24 and 23" do you mean the WRF-simulation here?*

**Authors'answer:** Yes we mean WRF-simulation. This has been corrected.

***P19-L14*** *Do you mean -6 and -7 Wm-2? These figures represent fairly small losses compared to mid-latitude sites (e.g. -25 to -20 Wm-2 in Giesen et al, 2009), which is presumably due to the cold surface temperature of the snow surfaces.*
*Giesen RH, Andreassen LM, Van den Broeke MR and Oerlemans J (2009) Comparison of the meteorology and surface energy balance at Storbreen and Midtdalsbreen, two glaciers in southern Norway. Cryosphere, 3(1), 57–74, doi: 10.5194/tc-3-57-2009*

**Authors'answer:** Yes, we mean "-6 and -7 W m$^{-2}$" and this has been corrected. The reference to the study made by Giesen et al., has been added to address the comment above. The text now reads:
"The contribution of net LW on the other hand is low for all simulations (annual mean of -7/-6 W.m$^{-2}$ for AWS/WRF-simulations respectively). Note that these losses are small in comparison to mid-latitude sites (e.g. -25 to -20 W m$^{-2}$ according to the study by Giesen et al, (2009)), because of the very dry conditions of the atmosphere and to the cold surface temperature of the snow surfaces."

***P22-L1*** *Are these figures monthly average amounts for only snow-covered grid cells or something else? Please explain the averaging procedure in the methods section.*

**Authors'answer:** Yes the mean calculation only consider snow-covered grid cells. A sentence has been added in the methods section to clarify this: "Note that sublimation and energy balance are only computed over snow surfaces. This means that annual and monthly means are only computed at grid-cells with snow."

***P23-L1*** *Figure10 – the sublimation ratio and ablation rates do not seem to match up – the ratio at high elevations is ~100% which implies there is little melt, but for both AWS and WRF forcing there is still a significant melt rate, and for AWS in 2015, melt rate=sublimation rate. Is this an artefact of the averaging of melt rate over snow only? Also, why do melt rates increase with height? Perhaps you are better to present the ablation totals rather than the ablation rates? Either way, the top and bottom panels should be consistent.*

**Authors'answer:** Thank you for your comment, you were right; the ratios were computed incorrectly. This has been fixed and the figure has been changed accordingly:

[Figure]

*Figure 8: Simulated annual sublimation ratio (a,b) against elevation band using AWS (blue) and WRF (purple) forcing for 2014 (a,c) and 2015 (b,d). Simulated annual ablation ratio (sublimation and melt) against the elevation using AWS and WRF-forcing for 2014 (c) and 2015 (d).*

**P26-L30** *It is not clear what you mean by evaposublimation? Do you mean the evaporation of liquid from a melting surface or a combination of this process + sublimation from a frozen surface? If you are including evaporation from a melting surface, then doesn't the increased rate occur because the melting lowers the latent heat required to transform the water to vapour? Please be more explicit about the process occurring.*

**Authors'answer:** To avoid confusion this word has been removed. The sentence now reads: "This may be explained by warmer conditions which induce a warmer snow pack, increasing the saturated vapor pressure at the snow surface, providing energy to increase the sublimation rates (Herrero and Polo, 2016)"

**Figure S2** *If this precipitation sensitivity analysis is retained, Figure S2 needs to be included in the results section as it is discussed directly.*

**Authors'answer:** As the paper has been restructured we decided to keep this figure in the supplementary material.

---

## Author Comment (AC3) · 10 Jun 2019

Please see the attached supplement

Please also note the supplement to this comment:
https://www.the-cryosphere-discuss.net/tc-2019-31/tc-2019-31-AC3-supplement.pdf

---

## Author Comment (AC4) · 10 Jun 2019

Please see the attached supplement

Please also note the supplement to this comment:
https://www.the-cryosphere-discuss.net/tc-2019-31/tc-2019-31-AC4-supplement.pdf

---

## Author Comment (AC5) · 10 Jun 2019

Please see the attached supplement

Please also note the supplement to this comment:
https://www.the-cryosphere-discuss.net/tc-2019-31/tc-2019-31-AC5-supplement.pdf

---

## Author Comment (AC7) · 10 Jun 2019

**SUPPLEMENTARY MATERIAL**

**A – Climatology at La Laguna**

[Figure]

*Figure S1: (a) Monthly precipitation and (b) air temperature recorded at La Laguna station. The monthly mean*
*(black) is computed over the 1976-2016 period.*

**B – Comparison AWS vs. WRF grid point**

*Table S1: Statistics (R2, RMSE and AME (Absolute Mean Error)) between micromet outputs forced by the AWS*
*measurements and the WRF outputs.*

| | Vertical offset with the WRF grid point | T (°C) | | | RH | | | SWi | | | LWi | | | WS (m s⁻¹) | | | P (mm d⁻¹) | | |
|---|---|---|---|---|---|---|---|---|---|---|---|---|---|---|---|---|---|---|---|
| | | $R^2$ | RMSE | AME | $R^2$ | RMSE | AME | $R^2$ | RMSE | AME | $R^2$ | RMSE | AME | $R^2$ | RMSE | AME | $R^2$ | RMSE | AME |
| Colorado Bajo | 278m | 0.94 | 5.02 | 5.32 | 0.04 | 15.7 | 15.7 | 1 | 1.29 | 0.65 | 0.96 | 5.53 | 4.10 | 0.01 | 1.61 | 1.29 | 0.35 | 2.96 | 3.42 |
| La Laguna | 144m | 0.91 | 5.01 | 4.60 | 0.04 | 19.4 | 14.8 | 1 | 1.65 | 0.72 | 0.94 | 6.66 | 5.00 | 0.15 | 1.71 | 1.15 | 0.51 | 5.72 | 3.52 |
| Llanos de las Liebres | 305m | 0.91 | 6.39 | 6.06 | 0.04 | 20.6 | 15.8 | 1 | 1.34 | 0.51 | 0.98 | 5.57 | 4.48 | 0.37 | 2.08 | 1.46 | 0.51 | 7.17 | 4.31 |
| La Gloria | 56m | 0.89 | 5.33 | 4.81 | 0.08 | 19.7 | 15.2 | 0.99 | 2.95 | 1.09 | 0.93 | 6.61 | 5.12 | 0.17 | 1.34 | 1.01 | 0.58 | 5.05 | 3.05 |
| Colorado Alto | 196m | 0.89 | 6.61 | 6.27 | 0.04 | 21.6 | 16.6 | 1 | 1.54 | 0.53 | 0.99 | 4.46 | 3.69 | 0.52 | 2.31 | 1.70 | 0.51 | 8.35 | 4.16 |
| Vega Tapado | -145m | 0.88 | 7.01 | 6.73 | 0.03 | 22.8 | 17.6 | 0.99 | 3.43 | 1.40 | 0.99 | 4.24 | 3.67 | 0.53 | 4.28 | 3.48 | 0.53 | 8.95 | 4.73 |
| Tapado | 462m | 0.88 | 6.83 | 6.54 | 0..01 | 22.9 | 17.8 | 0.99 | 2.98 | 1.33 | 0.99 | 1.48 | 1.11 | 0.52 | 3.45 | 2.78 | 0.51 | 9.48 | 4.92 |
| Paso Aguas Negras | -121m | 0.83 | 7.29 | 6.99 | 0.00 | 24.6 | 19.0 | 1 | 1.96 | 0.75 | 0.98 | 2.96 | 2.39 | 0.54 | 5.19 | 4.27 | 0.56 | 7.89 | 4.25 |

10

**C-** Spatial variability of the annual SCD.

[Figure]

*Figure S2:* Snow cover duration from MODIS images for (a) 2014 and (d) 2015. Simulated snow cover duration using AWS-forcing for (b) 2014 and (e) 2015. Simulated snow cover duration using WRF-forcing (c) 2014 and (f) 2015.

10

15

**D- Influence of the precipitation amount on the energy fluxes contribution**

[Figure]

*Figure S3: Monthly average (for the entire catchment, over snow surfaces only) of the main modeled heat fluxes for (a) 2014 and (b) 2015 AWS-forcings, (c) a simulation using the 2015 precipitation and the 2014 AWS-forcings, and (d) a simulation using the 2014 precipitation and 2015 AWS-forcings. SW is net shortwave radiation, LW is net longwave radiation, QE is the latent heat flux and QH is the sensible heat flux.*

*Figure S2: Monthly simulated melt (purple) and sublimation (blue) using AWS-forcing (a,b) and AWS-forcing with (c) 2014 precipitation and (d) 2015 precipitation. Forcing. Back numbers indicate the monthly sublimation ratio in %.*

**E. Model uncertainties**

The main forcing uncertainties were estimated to be the precipitation (related to P measurements and the P spatialization as there are only 2 recording stations); and the wind speed spatialization (Gascoin *et al.,* 2013). Regarding the other variables T, RH, SW, and LW, a similar study performed in a nearby catchment indicated a very good performance of MicroMet to spatialize these variables, with high correlations and low biases (for more details please refer to Gascoin *et al.,* 2013).

The principal calibration uncertainties are the topographic length scale used for the wind distribution and *z0* due to the absence of measurements to properly calibrate the model. Note that the albedo measurements have been used to calibrate the model, that is why this calibration is not considered as a main calibration uncertainty in this

study. Indeed, if we compare the model output albedo with the measurements, the mean *R* is 0.74 with mean *RMSE* of 0.22 and errors are considered related to the simple approach of the albedo computation of the model.

*Table S2: Mean annual melt and sublimation ratio of each simulation. Bold values indicate when results are different than the reference study (i.e. results present in section 4 with the AWS-forcings) indicated in the first line.*

| | 2014 | | 2015 | |
|---|---|---|---|---|
| | Melt (m.w.e.) | Sub. (m w.e.) (Sub. Ratio) | Melt (m.w.e.) | Sub. (m w.e.) (Sub. Ratio) |
| Reference sim. | 0.047 | 0.030 (42%) | 0.21 | 0.19 (48%) |
| Precipitation +10% | **0.048** | **0.029 (41%)** | **0.23** | 0.19 **(47%)** |
| Curvature 100m | 0.047 | 0.030 (42%) | 0.21 | 0.19 (48%) |
| Curvature 1000m | 0.047 | 0.030 (42%) | 0.21 | 0.19 (48%) |
| Slope 0.25 - Curv. 0.75 | 0.047 | 0.030 (42%) | 0.21 | 0.19 (48%) |
| Slope 0.75 - Curv. 0.25 | 0.047 | 0.030 (42%) | 0.21 | 0.19 (48%) |
| *Z0* = 10 mm | **0.033** | **0.049 (66%)** | **0.12** | **0.29 (72%)** |

**E1. Precipitation uncertainties**

**a) Precipitation measurements**

As mentioned in section 5.1.2 snowfall measurements in windy conditions can suffer from an undercatch bias, and corrections are generally performed using empirical relationship (e.g. MacDonald and Pomeroy, 2007; Wolff *et al.,* 2015). The strong wind gusts in this region, and especially at high elevations (Figure 3) increase measurement uncertainties. Nevertheless, in this study it was chosen not to apply any correction. First, given the lack of continuous SWE measurements, it is not straightforward to establish an empirical correction. Also, precipitation data from two rain gauges (one shielded and the second one unshielded) located in Tapado have been averaged, to reduce the random error. Note that the data from the two rain gauges is surprisingly similar (see section 2.2.1), while we would expect a larger amount of precipitation catched by the shielded Geonor sensor, in this area of strong wind speed. This suggest that either the under catch is not si important or that the shield is not that efficient.

**b) Spatialization**

The main uncertainty for the precipitation data is due to the data spatialization at catchment scale. As only two stations have been used to force the model, the uncertainty is expect to be significant due to the orographic complexity of the catchment, but cannot be evaluated based on the current available measurements.

The interpolation of precipitation in MicroMet subroutine has been done without the use of an altitudinal gradient (called precipitation adjustment factor, Liston and Elder, 2006a) as no consistent altitudinal gradients were found in the observations (results not shown). Each event is very specific (e.g. Sinclair and MacDonell, 2016) and as a result, the altitudinal lapse rate can be either positive or negative. In addition, due to low precipitation rate and the few number of events, mainly in 2014, most of the time precipitations are recorded at one station but not at the other, or there is a delay between the events recorded at both stations, which are 14 km from one another. Different altitudinal gradients (monthly *vs* event scale) were tried but the best comparisons between simulated and observed snow depths were found without considering any gradient. Considering an altitudinal lapse rate systematically leads to an over-estimation, especially at high elevations, where the snow persists until the next season.

In addition, the model simulates the wind transport only after deposition and does not consider the preferential deposition. Again, the available measurements make it difficult to estimate this uncertainty.

**c) Sensitivity test**

A sensitivity test was performed by increasing the precipitation by 10%. Results indicate that the mean annual sublimation ratio over the catchment is very similar to that of the initial simulation (Figure 2). Indeed, it decreases by 1% for 2014 and for 2015. Regarding the spatial variability (Figure 11) the difference varies between +1 and -3 %, and larger differences are observed for 2014 between 5000 and 5500 m a.s.l. and around 4600 m a.s.l. for 2015. This maximum difference can be related to changes in spatial distribution of snow when some events occur only at high elevations.

**d) Impact on model performances**

The precipitation uncertainties explain also the better performances of the model for 2015 than 2014. Indeed, the model is sensitive to patchy snow, mainly observed at low elevation. As mentioned in section 4.2.1, modeling the spatial variability of the SD is complex and results show an overall similarity of the simulated SD between some stations. When dealing with low amount of snow, patches over the catchment are more present an increase the error.

**E2. Wind uncertainties**

**a) Wind data spatialization uncertainty**

As mentioned by Gascoin *et al.,* (2013), the wind speed simulated by MicroMet model tend to be under-estimated especially at low elevation. With the exception of Paso del Agua Negra, similar results are found in this study, with largest bias found at low elevations. Indeed, the results of the cross validation test indicate an *RMSE* of at

4.1 m.s$^{-1}$ at La Laguna and a larger bias (RMSE = 7.8 m.s$^{-1}$ at Paso del Agua Negra. Differences are related to the MicroMet interpolation and can be explained by different reasons. First, The main shortcoming of the wind interpolation module is the lack of thermal winds (e.g. katabatic winds). In addition MicroMet does not take any topography into account that determines the dynamic wind direction, implying bias in the wind interpolation.

5   Nevertheless, the absence of general trends of the under-estimation (both low and high values are under-estimated and the bias strongly depend on the location and the wind speed) makes difficult to establish a relationship to correct the bias and also to evaluate the uncertainty at catchment scale.

Therefore, the number of stations used is important as increasing the number will decrease the uncertainty. The wind uncertainty has an impact on the sublimation ratio. Indeed, for instance, increasing the wind speed by 10%

10   can induces sublimation ratio changes of 40% at high elevation (i.e. where there is no melt). The wind speed is, in fact, known to directly affect the turbulent fluxes (e.g. Litt et al., 2017)

In addition, the wind speed also impacts the snow density, which directly affects the thermal conductivity of the upper snow layers (Yen, 1981). As a consequence it impacts the surface temperatures thus the turbulent fluxes.

**b) Topographic influence on wind transport**

15   In the model, the curvature length and the influence of the slope *vs.* curvature can be calibrated to consider the topographic influence of the wind transport. While the curvature length can be approximately constrained from the DEM, the relative influence of slope and curvature is more difficult to quantify a priori. As such, sensitivity tests have been performed to evaluate the impact of these parameters on the simulation.

First, the curvature length value has been varied from 100m to 1000m, and does not impact the annual mean

20   sublimation ratio at the catchment scale (Figure 2). Nevertheless the spatial variability of the differences depends on the elevation (Figure 11). Considering a larger curvature length leads to larger sublimation ratio at low elevation (in the valley) and lower one at high elevation. When choosing a larger curvature value, the simulated snow depth is larger in the valley as the snow transport is from a larger area, and increasing the snow depth decreases the sublimation ratio. Indeed, in that case the snow persists longer in spring, and warmer temperature

25   allow increasing the melt rate.

The annual mean of the sublimation ratio at the catchment scale is not sensitive to the influence of the slope *vs.* curvature either (Figure 2) when values are ranging between 0.25 and 0.75. Nevertheless, varying this influence changes the spatial distribution of snow depth. Larger snow depths (between 18 and 26%) are observed on the ridges when the influence of the slope is larger than the curvature (results not shown). As a consequence, the

30   sublimation ratio decreases by about 10 to 20% in areas with thicker snow, and can also be explained by a the persistence of the snow cover during the spring, increasing the melt rate. On the contrary the sublimation ratio is

larger on steep slopes, when the influence of the slope is set to be larger than the curvature, as this calibration allows for more snow redistribution, decreasing the mean snow depth.

The change in snow distribution is more important when changing the influence of slope vs. curvature from 0.25 to 0.75 than when changing the curvature length from 100 to 1000 m. The calibration of these parameters remains important when the sublimation is evaluated in the valley for instance but according to our results, it does not affect the sublimation ratio when evaluated at the catchment scale

**E3. Roughness value**

Increasing the roughness value by a factor 10 increases the annual mean of the sublimation ratio by 24% for 2014 and by 20% for 2015 (Table S1). Larger changes are observed around 4600 m a.s.l. for 2014 and 2015 where the difference can reach 30% (Figure 11) .

This sensitivity test highlights the strong sensitivity and the importance of choosing an accurate value to properly quantify sublimation over the catchment. Further studies are therefore recommended to calibrate this value using turbulent flux measurements such as with an Eddy Covariance System (e.g. Litt *et al.,* 2017). Nevertheless even with measurements, significant uncertainty remains due to the strong spatio-temporal variability of the snow surface roughness. While the roughness value is used as a calibrated parameter and is not an absolute physical value, it depends on the surface roughness. The roughness is expected to increase with elevation, as penitentes are commonly observed on the lower Tapado glacier (e.g. Lhermitte et al., 2014; Nicholson et al., 2016) and surrounding areas. There is therefore a strong spatial variability of the roughness value, as penitentes are not observed over the entire catchment, but mainly in the upper part. In addition, penitentes grow in size over the season (Lliboutry, 1954) leading to a strong temporal variability of the roughness.

Due to the strong sensitivity to $z_0$ and the potential for significant spatio-temporal variability of the snow surface roughness, to reduce uncertainties, a spatio-temporal evolution of $z_0$ could be envisaged. At this stage, without more measurement it is a complicated task. Further studies based on two EC measurements over the season could help to evaluate the variability.

[Figure]

*Figure S4: Differences in sublimation ratio per elevation range for (a) 2014 and (b) 2015, between: reference simulation and the simulation performed with a precipitation increase at the two AWS by 10% (blue); reference simulation ($z_0$=1mm) and with an increase of $z_0$ to 10mm (green); 100 and 1000m curvature length simulations (orange) and slope vs. curvature weight of 0.25 – 0.75 and 0.75 – 0.25 (black).*

---

## Referee Report (RR1)

Réveillet et al. have restructured the manuscript with the new title "Impact of forcing on sublimation simulations for a high mountain catchment in the semi-arid Andes" fundamentally with respect to the reviewer's comments. The focus on the differences in sublimation depending on the forcing by AWS data or WRF model output is presented much more comprehensively in the updated version.

In general, the manuscript can be optimized by highlighting and rating the most important parameters, their representation in the different products (AWS, WRF) and their impact on processes controlling sublimation rates in the final conclusions. In addition, the advantage of using WRF data (I assume the application in forecast) should be pointed out with respect to the presented outlook.

The manuscript is worth for publishing after corrections according to the comments hereafter.

Specific Comments

(1) The snow depth (SD) in Fig. 3 is presented in m w.e. First, I assume that this is SD in meter, and not the SWE. In section 4.2 the RMSE error is given by 0.15 m. This is about the same magnitude as the total snow depth in 2014, and considerable in relation to SD in 2015. Please discuss the ration of RMSE to total snow depth in order to rate the overall performance of SD simulations.

(2) Within this study multiple parameters and processes are shown influencing the simulated sublimation. In addition to the high uncertainty in precipitation (and thus SD, SCA, SCD) also the roughness value is shown to have a high impact on results. Likewise, the ground level wind can be assumed to control turbulent heat fluxes considerably. On page28 line 24 several sources of uncertainty are mentioned. I would like to encourage the authors to present on the basis of the results and the discussion a final ranking of the largest uncertainties and their impact on simulated sublimation in the conclusion. Please also state which potential and advantages are expected by improving WRF using AWS data assimilation vs using MicroNet interpolation of AWS data (re-analysis or fore-cast? see last sentence (iii)).

P1 L21: To complete the information in the brackets add the catchment size

P1 L25: Here and throughout the text: please try to avoid the slash (here: melt/sublimation). It might be confusing if this is "and" or "or". In this case it also can be read as the ratio of melt divided by sublimation.

P1 L28: Here more detailed information on the processes (SW radiation, sensible heat flux, …) causing higher melt rates would be desirable.

P1 L29: One sentence might be added on the overall applicability of WRF output for forcing the snow-pack simulation.

P5 L9: Do you have experience or any data on how the temperature and RH measurements of the HOBO weather stations perform in comparison to the permanent AWS. Are the HOBO temperature sensors ventilated? Please add one or two sentences.

P12 L18: AWS 2014 vs 2015

P12 L24: Replace "larger number of clouds" by "higher degree of cloud cover"

P12 L30: Please present the relative difference in addition to the absolute value. Please also state that this sums up to a factor of three for highest elevations (see Fig. 4d).

P14 L1: What is the temperature along the precipitation event on June 21$^{st}$ 2015? Can the temperature threshold between rain and snow be also a reason for the overestimation of snow depth?

P16 L12ff and Fig. 5: Please discuss possible reasons for this high variation of SCA in the text.

P20 L9: For 2015 the modelled turbulent fluxes…

P24 L15: Please mention here why it would be advantageous to correct the regional WRF model output in contrast to use the MicroMet interpolation of station values.

P27 L13: Referring to the title, sublimation rates have been the specific focus. Of cause do snow depth and SCA play an important role since they result from precipitation, which is highly uncertain (see conclusions).

P27 L15: Since sublimation is most sensitive to the roughness value, wind at the ground level also plays an important role once a snow cover exists. Please add this information.

P28 L15: Delete "years" at the end of the sentence.

Figure 3:

- It's not clarified in the caption if hourly wind speed or daily mean values of wind speed are presented. I would suggest to present hourly values to enable a better estimate of potential wind transport and sublimation rates from the graph.

Figure 10

- Caption: Revise "Cumulated" by "Stacked".

---

## Referee Report (RR2)

Review of revised manuscript "Impact of forcing on sublimation simulations for a high mountain catchment in the semi-arid Andes" by Réveillet et al.

Jono Conway

**General comments**

The manuscript is much improved from the initial submission. The paper now focusses more on the differences in sublimation when using different forcing datasets, with a smaller focus on differences in sublimation due to interannual climate variability and elevation. The new and amended figures highlight useful information and the results have been structured in a logical fashion.

The interpretation and discussion around the central result of the paper (increased sublimation rate and ratio when using WRF output) is, however, still lacking and deserves much more attention. The assertion that differences in snow cover are driving the differences is not compelling and is not supported by all the results. For instance, it does not explain why sublimation is still higher in 2015 when the snow-covered area is similar between WRF and AWS simulations. Colder temperature and lower RH will tend to favour sublimation over melt, but these factors do not explain the larger magnitude of both the sensible and latent heat for WRF simulations across all elevations in both years.

I would suggest that wind speed is one of the key reasons for differences in sublimation that needs discussion. Figure 4 shows large differences in wind speed between AWS and WRF forcing (average ~2 vs ~5 ms-1). The turbulent fluxes are very sensitive to variations in wind speed in this range, especially if stability corrections are employed. Figure 9 shows both latent and sensible heat fluxes are increased in WRF simulations compared to AWS simulations, indicating that wind speed is driving an increased magnitude of turbulent fluxes (and hence sublimation). Another piece of evidence that wind speed is driving increased sublimation is the increased sublimation in 2015 for AWS simulations, which have higher wind speed than in 2014.

The interpretation in the results and discussion (including sections 4.3.2, 5.1, 5.1.2 and specific comments) needs revised to align with the new results presented.

In addition, some of the methods that are central to the key results need further description. In particular: the calculation of turbulent fluxes, extrapolation of radiation, definition of mass balance terms, validation of surface temperature (see specific comments).

One further area of concern is the discrepancy between the total precipitation and total ablation simulated by each forcing dataset. In 2015, snow covered area starts and ends the year close to 0 (Figure 6) implying that all the snow that fell during the year has been lost in the year. Both forcing datasets result in a similar magnitudes of SCD and ablation rate with elevation (figures 7 and 8) as well as catchment average total ablation (figure 10), which appears to be around 0.4 m w.e.. Yet the precipitation in WRF is between 0.20 to 1 m w.e. larger than the AWS forcing. What has happened to this extra precipitation in the WRF simulations? Has it fallen as rain? Please explain.

Once these points are addressed the paper will make a valuable contribution to the literature.

**Specific comments:**

P1ln24 "*melt/sublimation ratios*" Please be consistent with the terms used in the rest of the manuscript i.e. "ratio of sublimation to total ablation (sublimation+melt)"

P1ln25 "*due to…*" This statement needs revising in line with the major comment above. It is not clear that the increased precipitation increases the sublimation ratio. In fact, this is at odds with the last sentence in the abstract.

P8In10 Given the overestimation of wind speed in the WRF output, it would be worth checking that the data used were in fact supplied at 2m height - standard WRF outputs for surface wind speed are given at 10m and will be markedly higher than 2m wind speed.

Section 3.1 or 3.2 Please describe in more detail the mass balance terms used. I.e. does 'sublimation' include evaporation from a melting surface? Does melt include melt that is refrozen in the snowpack or only that which drains from the snowpack? This will make a big difference to the sublimation ratio.

P10In30 Please describe in more detail the relevant aspects on the turbulent flux parameterisation in SnowModel (i.e. surface temperature calculation, stability correction, whether the option to enhance turbulent fluxes for patchy snow was used etc). This is a key aspect of the methods the deserves more attention.

P12In30 The variation of LW and SW with elevation in Figure 4f,g does not make sense - if Ta and RH are lower with WRF, then LW should be less but the WRF values are very similar to the AWS values in each year. Similarly, the SW values for WRF and AWS data are very close for each year, despite larger differences in RH between WRF and AWS than between each year for each data source. Can the authors check the lines are labelled correctly and revise comments regarding differences in LW and SW.

Following on from the above point, section 3.2.1 needs to describe (at least in general) how short and longwave fluxes are extrapolated. From reading Liston and it seems that cloud cover is estimated using RH, then predetermined cloud extinction coefficients are used to calculate SW and LW across elevations. Was the option to 'assimilate' the observations used in this study? Neither of these is particularly standard practice when SW and LW observations are available to distribute LW and SW across a catchment, and these inputs will have a large bearing on the simulated energy and mass balances.

P13In10 It is good to see the validation of the surface temperature (albeit at one site only), and it would be useful to include this at the beginning of section 4.2 or at least the supplementary material, given that a correct estimation of surface temperature is key to a correct simulation of melt vs sublimation. The surface temperature simulated with WRF should also be included on this figure too – I expect it will be lower than the AWS measurements given the predominance of sublimation.

P22In6 "This [larger turbulent fluxes in WRF vs AWS for 2014] can be explained by the larger snow cover simulated by WRF-forcing with snow cover in the entire catchment while the AWS-forcing only results in snow at higher elevations. Since the fluxes are computed over snow surfaces only, WRF increases the contribution of warmer, low elevation areas." Figure 8 shows that the differences in turbulent fluxes are similar at all elevations, which does not support this statement. In 2015, the snow cover is similar between WRF and AWS simulations, yet turbulent fluxes and sublimation are still much larger in the WRF simulations – suggesting that snow cover duration is not driving the differences in sublimation. Please remove or revise this statement.

P24In26 "On the other hand biases in wind speed, incoming LW and SWi and air pressure are low for both years (results not shown)." Figure 4 now wind speed, LWi and SWi, this comment needs revised. A large bias in wind speed is shown in the figure and needs discussing.

P25I6 "Snow that persists over a longer period results in an increased melt rate and can influence the sublimation rate and ratio (especially in 2015, Figure 10)." This comment is confusing as SCD in 2015 was similar between AWS and WRF simulations, hence it cannot be the driver for differences between WRF and AWS. Please revise.

P2518 "This is the only explanation for the larger melt rate observed with AWS-forcing compared to WRF-forcing at low elevations, given the cold bias in WRF-forcing (Figures 8c and 4b)." Figure 10c shows that in 2014 the low elevation areas show a similar melt rate between WRF and AWS simulations so it is unclear what is meant here. The cold bias in WRF forcing would tend to favour lower melt, which would explain the higher melt with AWS forcing. Please revise.

**Editorial comments:**

Figure 4f,g – please check the lines are labelled correctly.

Figure 5a – should the legend read "AWS sim" rather than "WRF sim"

Figure 5 caption – correction to colours – "vs. observed (red)..." and "Grey shaded areas.."

Figure 8a ylabel – "...(%) for 2014"

Figure 8 caption – "... forcing for 2014 (a) and 2015 (b). Simulated annual average total ablation (sublimation and melt) ...

Figure 9 – common x limits would make it easier to compare between years as well as with elevation.

Figure 11 – common y limits would make it easier to compare between forcing datasets.

---

## Author Response (AR2)

**Response to reviewer #1**

*1- General Comments*

*Réveillet et al. have restructured the manuscript with the new title "Impact of forcing on sublimation simulations for a high mountain catchment in the semi-arid Andes" fundamentally with respect to the reviewer's comments. The focus on the differences in sublimation depending on the forcing by AWS data or WRF model output is presented much more comprehensively in the updated version.*

*In general, the manuscript can be optimized by highlighting and rating the most important parameters, their representation in the different products (AWS, WRF) and their impact on processes controlling sublimation rates in the final conclusions. In addition, the advantage of using WRF data (I assume the application in forecast) should be pointed out with respect to the presented outlook.*

*The manuscript is worth for publishing after corrections according to the comments hereafter.*

**Authors' response:** Thanks again to the reviewer for new insightful comments that have helped to strengthen the final paper.

*2- Specific Comments*

*(1) The snow depth (SD) in Fig. 3 is presented in m w.e. First, I assume that this is SD in meter, and not the SWE. In section 4.2 the RMSE error is given by 0.15 m. This is about the same magnitude as the total snow depth in 2014, and considerable in relation to SD in 2015. Please discuss the ration of RMSE to total snow depth in order to rate the overall performance of SD simulations.*

**Authors' response:** Thank you for pointing out this mistake, m w.e. have been changed to m.

The RMSE mentioned in section 4.2 corresponds to the mean of the RMSE computed at each station. When forced by the AWS-forcing, it ranges between 0.06 and 0.19 m w.e. (indicated in Figure 5) corresponding to 11 and 26 % of the maximum snow depth at each station. In particular large RMSE computed for 2014 corresponds to 63 %. Details are reported in the Table below and the % is now given in section 4.2:

Simulated snow depths using the AWS-forcing (Figures 5 a-f) are in good agreement with measured snow depth values (mean $k$=0.14 and mean $RMSE$=0.15 m, corresponding to 36% of the maximum mean snow depth). Note that the largest RMSE corresponds to 63% of the maximum snow depth.

[…]

Simulations performed with the WRF-forcing indicate lower performances in simulating snow depth evolution at the AWS (Figure 5 g-m; mean $k$=0.12, and mean $RMSE$=0.20 m, corresponding to 39% of the maximum mean snow depth, and the largest RMSE corresponds to 76 % of the maximum snow depth).

| AWS | Max Snow measured (m) | RMSE AWS-forcing | % | RMSE WRF-forcing | % |
|---|---|---|---|---|---|
| La Gloria | 0.45 | 0.14 | 31% | 0.34 | 76% |
| Vega Tapado | 0.35 | 0.17 | 59% | 0.16 | 27% |
| Colorado Alto | 0.35 | 0.18 | 63% | 0.15 | 43% |
| La Laguna | 0.59 | 0.17 | 28% | 0.30 | 51% |
| Tapado | 0.56 | 0.06 | 11% | 0.07 | 13% |
| Tapado Alto | 0.74 | 0.19 | 26% | 0.19 | 26% |
| Mean | 0.51 | 0.15 | 36% | 0.20 | 39% |

*(2) Within this study multiple parameters and processes are shown influencing the simulated sublimation. In addition to the high uncertainty in precipitation (and thus SD, SCA, SCD) also the roughness value is shown to have a high impact on results. Likewise, the ground level wind can be assumed to control turbulent heat fluxes considerably. On page28 line 24 several sources of uncertainty are mentioned. I would like to encourage the authors to present on the basis of the results and the discussion a final ranking of the largest uncertainties and their impact on simulated sublimation in the conclusion. Please also state which potential and advantages are expected by improving WRF using AWS data assimilation vs using MicroNet interpolation of AWS data (re-analysis or fore-cast? see last sentence (iii)).*

**Authors' response:**

More information are now given in the conclusion, according to your comment. It now reads:

"Sublimation simulated in this study is associated with several sources of uncertainty related to the forcing chosen and the model calibration. Regarding the calibration, the roughness value is the key concern to properly simulate the turbulent fluxes and result showed strong sensitivity to this value. Nevertheless, without measurements to properly calibrate this value, it appears to be the main source of model calibration uncertainty. Results presented here highlight precipitation as the main forcing uncertainty, due to measurement errors and lack of spatial representation as precipitation data was only available for two stations. Precipitation uncertainties directly impact snow on the ground, and therefore indirectly sublimation rates. Uncertainties in wind speed were likely the second source of error in sublimation results, which needs to be better constrained in future studies. Therefore, this study highlights that this uncertainty has a strong impact on sublimation and further work is suggested to *(i)* improve measurements uncertainties, *(ii)* increase the number of sensors over the catchment, and *(iii)* incorporate AWS measurements into the WRF model and use data assimilation to improve model outputs. CEAZA is currently working on point *(iii)* to provide improved WRF outputs for the semi-arid Andes of Chile. This study has highlighted the current difficulties in using standard WRF model outputs in a semiarid, Andean catchment. Moving forward, it would be highly advantageous to improve WRF model performance in mountainous areas where high relief and difficult access often limits AWS distribution to valley floors, therein limiting the accuracy of interpolation techniques for terrain sensitive variables, such as precipitation and wind speed and direction."

*3- Detailed Comments (P = page, L = line)*

*P1 L21: To complete the information in the brackets add the catchment size*

**Authors' response:** Done:

"…this study aims to simulate melt and sublimation rates over the instrumented watershed of La Laguna (513 km$^2$, 3150–5630 m a.s.l., 30°S 70°W), during two hydrologically contrasting years (i.e. dry *vs*. wet)."

*P1 L25: Here and throughout the text: please try to avoid the slash (here: melt/sublimation). It might be confusing if this is "and" or "or". In this case it also can be read as the ratio of melt divided by sublimation.*

*P1 L28: Here more detailed information on the processes (SW radiation, sensible heat flux, ...) causing higher melt rates would be desirable.*

*P1 L29: One sentence might be added on the overall applicability of WRF output for forcing the snow-pack simulation.*

"Results of simulations indicate first a large uncertainty in sublimation to melt ratios depending on the forcing as the WRF data has a cold bias and over-estimates precipitation in this region. These input differences cause a doubling of the sublimation to melt ratio using WRF forcing inputs compared to AWS. Therefore, the use of WRF model output in such environments must be carefully adjusted so as to reduce errors caused by inherent bias in the model data. For both input datasets, the simulations indicate similar sublimation fraction for both study years, but ratios of sublimation to melt vary with elevation as melt rates decrease with elevation due to decreasing temperatures. Finally results indicate that snow persistence during the spring period decreases the ratio of sublimation due to higher melt rates."

*P5 L9: Do you have experience or any data on how the temperature and RH measurements of the HOBO weather stations perform in comparison to the permanent AWS. Are the HOBO temperature sensors ventilated? Please add one or two sentences.*
**Authors' response:** No direct comparison data are available in the study area. The temperature sensors are not ventilated, however relatively constant wind allows natural ventilation of the sensors. We agree that the HOBO's sensors performances are lower than for Campbell Scientific sensors and have added a sentence to indicate this in the text (se below). Nevertheless, we still think that the main source of error is from the interpolation more than the measurements.
"Although the lower accuracy of HOBO weather sensors compared to the permanent stations represent a source off errors in the forcing data, errors resulting from the spatial interpolation of forcings are likely to be much greater than these."

*P12 L18: AWS 2014 vs 2015*
**Authors' response:** Done

*P12 L24: Replace "larger number of clouds" by "higher degree of cloud cover"*
**Authors' response:** Done

*P12 L30: Please present the relative difference in addition to the absolute value. Please also state that this sums up to a factor of three for highest elevations (see Fig. 4d).*
**Authors' response:** This information has been added, according to your comment:
"and higher precipitation (annual cumulative difference larger than 1 m w.e. and a difference ranging between a factor of 1.6 to 3.4 depending on the elevation (Figure 4d))."

*P14 L1: What is the temperature along the precipitation event on June 21st 2015? Can the temperature threshold between rain and snow be also a reason for the overestimation of snow depth?*
**Authors' response:** It is unlikely that rainfall causes the difference, as no rainfall has been observed, or calculated, in the catchment. The overestimation of snow depth is likely due to the results of the interpolation calculation, which is based on relatively low snowfall rates at the La Laguna AWS compared with higher elevations. We have further clarified this point in the text.
"Interpolation results in overestimation of the simulated snow depth during 2015, probably due to an over-estimation of the precipitation for the large event on June 21st 2015 (Figure 3) caused by large differences in measured precipitation at the La Laguna and Tapado AWSs especially. Nevertheless, the start and the end date of the snow season are in good agreement with observations (maximum difference of 3 days observed at La Gloria site). Note that this comparison probably over-estimates the accuracy as snow depths are compared at the exact location of meteorological forcing. Larger uncertainties are expected at the interpolated locations."

**Authors' response:** We have added the following explanation to the Discussion (Section 5.1.2, p23-24):
The large variation of the SCA resulting from the WRF driven model results (Figure 5) is likely related to the higher frequency of relatively small precipitation events modeled by WRF than are recorded by the AWS. These small events cover the catchment with a relatively thin layer of fresh snow, which sublimates relatively quickly causing the SWE to decrease to < 3 mm w.e. at lower elevations, causing high variability in modeled SCA.

*P20 L9: For 2015 the modelled turbulent fluxes…*
**Authors' response:** Done

*P24 L15: Please mention here why it would be advantageous to correct the regional WRF model output in contrast to use the MicroMet interpolation of station values.*
**Authors' response:** The following text has been added to the end of the Conclusion (Section 6):
"This study has highlighted the current difficulties in using standard WRF model outputs in a semiarid, Andean catchment. Moving forward, it would be highly advantageous to improve WRF model performance in mountainous areas where high relief and difficult access often limits AWS distribution to valley floors, therein limiting the accuracy of interpolation techniques for terrain sensitive variables, such as precipitation and wind speed and direction. "

*P27 L13: Referring to the title, sublimation rates have been the specific focus. Of course the snow depth and SCA do play an important role as they result from precipitation, which is highly uncertain (see conclusions).*
**Authors' response:** The first sentence now reads: The main objective of this study was to investigate the impact of forcing data on modeled mass and energy balance to explain sublimation ratios.

*P27 L15: Since sublimation is most sensitive to the roughness value, wind at the ground level also plays an important role once a snow cover exists. Please add this information.*
**Authors' response:** According to your comment, and in agreement with reviewer #2, a discussion about the role of the wind has been added. See response to Reviewer #2.

*P28 L15: Delete "years" at the end of the sentence.*
**Authors' response:** Done

*Figure 3:*
*- It's not clarified in the caption if hourly wind speed or daily mean values of wind speed are presented. I would suggest to present hourly values to enable a better estimate of potential wind transport and sublimation rates from the graph.*
**Authors' response:** The wind rose are result of combining hourly wind speed and velocity data. The word 'hourly' has been added to the caption.

*Figure 10*
*- Caption: Revise "Cumulated" by "Stacked".*
**Authors' response:** Done.

**Response to reviewer #2**

**1- General comments**

*The manuscript is much improved from the initial submission. The paper now focuses more on the differences in sublimation when using different forcing datasets, with a smaller focus on differences in sublimation due to interannual climate variability and elevation. The new and amended figures highlight useful information and the results have been structured in a logical fashion.*
*The interpretation and discussion around the central result of the paper (increased sublimation rate and ratio when using WRF output) is, however, still lacking and deserves much more attention. The assertion that differences in snow cover are driving the differences is not compelling and is not supported by all the results. For instance, it does not explain why sublimation is still higher in 2015 when the snow-covered area is similar between WRF and AWS simulations. Colder temperature and lower RH will tend to favour sublimation over melt, but these factors do not explain the larger magnitude of both the sensible and latent heat for WRF simulations across all elevations in both years.*
*I would suggest that wind speed is one of the key reasons for differences in sublimation that needs discussion. Figure 4 shows large differences in wind speed between AWS and WRF forcing (average ~2 vs ~5 ms-1). The turbulent fluxes are very sensitive to variations in wind speed in this range, especially if stability corrections are employed. Figure 9 shows both latent and sensible heat fluxes are increased in WRF simulations compared to AWS simulations, indicating that wind speed is driving an increased magnitude of turbulent fluxes (and hence sublimation). Another piece of evidence that wind speed is driving increased sublimation is the increased sublimation in 2015 for AWS simulations, which have higher wind speed than in 2014.*
*The interpretation in the results and discussion (including sections 4.3.2, 5.1, 5.1.2 and specific comments) needs revised to align with the new results presented.*
*In addition, some of the methods that are central to the key results need further description. In particular: the calculation of turbulent fluxes, extrapolation of radiation, definition of mass balance terms, validation of surface temperature (see specific comments).*
**Authors' response:** Thanks again for new insightful comments that have helped to strengthen the final paper. Response of the comments mentioned here are detailed below.

*One further area of concern is the discrepancy between the total precipitation and total ablation simulated by each forcing dataset. In 2015, snow covered area starts and ends the year close to 0 (Figure 6) implying that all the snow that fell during the year has been lost in the year. Both forcing datasets result in a similar magnitudes of SCD and ablation rate with elevation (figures 7 and 8) as well as catchment average total ablation (figure 10), which appears to be around 0.4 m w.e.. Yet the precipitation in WRF is between 0.20 to 1 m w.e. larger than the AWS forcing. What has happened to this extra precipitation in the WRF simulations? Has it fallen as rain? Please explain.*
**Authors' response:** There is no extra precipitation with regards to ablation results, there is a zero sum for all modeled years for both WRF and AWS driven outputs. The confusion may lie with the graphical representation, which does not include relative percentage of the catchment. For example, in Figure 8, if the elevation is summed directly, it appears that the ablation rates for 2015 are very similar, however when summed over the snow-covered area they equal precipitation amounts (Table R1).

Table R1: Total values (m$^3$ w.e.) calculated in each simulation over the entire catchment

|  | 2014 | | | | 2015 | | | |
|--|-------------|------|-------------|------------------------|-------------|---------|-------------|------------------------|
|  | Precipitation | Melt | Sublimation | Precipitation-Ablation | Precipitation | Melt | Sublimation | Precipitation-ablation |
| AWS | 4174.5 | 3013.3 | 1161.2 | 0 | 22775.4 | 12167.5 | 10608.4 | 0 |
| WRF | 24354.4 | 4315.5 | 20038.9 | 0 | 54509.9 | 16046.4 | 38463.5 | 0 |

**2- Specific comments:**

*P1ln24 "melt/sublimation ratios" Please be consistent with the terms used in the rest of the manuscript i.e. "ratio of sublimation to total ablation (sublimation+melt)"*
*P1ln25 "due to…" This statement needs revising in line with the major comment above. It is not clear that the increased precipitation increases the sublimation ratio. In fact, this is at odds with the last sentence in the abstract.*
**Authors' response:** Corrected, please see response to reviewer 1. The new text is:
"Results of simulations indicate first a large uncertainty in sublimation to melt ratios depending on the forcing as the WRF data has a cold bias and over-estimates precipitation in this region. These input differences cause a doubling of the sublimation to melt ratio using WRF forcing inputs compared to AWS. Therefore, the use of WRF model output in such environments must be carefully adjusted so as to reduce errors caused by inherent bias in the model data. For both input datasets, the simulations indicate similar sublimation fraction for both study years, but ratios of sublimation to melt vary with elevation as melt rates decrease with elevation due to decreasing temperatures. Finally results indicate that snow persistence during the spring period decreases the ratio of sublimation due to higher melt rates."

*P8ln10 Given the overestimation of wind speed in the WRF output, it would be worth checking that the data used were in fact supplied at 2m height - standard WRF outputs for surface wind speed are given at 10m and will be markedly higher than 2m wind speed.*
Authors' response: We can confirm that the 10 m WRF output were logarithmically-scaled to the 2 m height before inclusion into MicroMet, similar to the way the data from the La Laguna AWS was treated (scaled from 10 m to 2 m height). The text now reads "The model outputs are at 2 m above the surface (note that wind output was logarithmically scaled from 10 m to 2 m height) and are available at 22 km resolution over Chile, 7 km resolution at the regional scale, and 3 km resolution over the La Laguna catchment (Figure 1)."

*Section 3.1 or 3.2 Please describe in more detail the mass balance terms used. I.e. does 'sublimation' include evaporation from a melting surface? Does melt include melt that is refrozen in the snowpack or only that which drains from the snowpack? This will make a big difference to the sublimation ratio.*
**Authors' response:** Further clarification have been added, and the edited text in Section 3.2 now reads:
"After validating the model, the sublimation ratio and sublimation and melt rates were computed over the catchment for the two years. The sublimation rate corresponds to the mass sublimated per unit of time and does not include evaporation from meltwater. The sublimation ratio is defined as a percentage, and equal to the sublimation divided by the total ablation (i.e. sublimation plus melt rates). The melt rate corresponds to meltwater that runs off from the snowpack. Whilst the model calculates refreezing, the final melt rate described here does not include snow melt that refreezes in

*P10ln30 Please describe in more detail the relevant aspects on the turbulent flux parameterisation in SnowModel (i.e. surface temperature calculation, stability correction, whether the option to enhance turbulent fluxes for patchy snow was used etc). This is a key aspect of the methods the deserves more attention.*

**Authors' response:** Unfortunately the model does not include the possibility to separately evaluate patchy snow. Regarding the other information, it has been added as follows:

"Note that the surface temperature is solved iteratively by closing the energy balance (Liston and Elder 2006). In addition, under stable atmospheric conditions, turbulent fluxes are modified based on a Richardson number correction (Liston and Hall, 1995)."

*P12ln30 The variation of LW and SW with elevation in Figure 4f,g does not make sense - if Ta and RH are lower with WRF, then LW should be less but the WRF values are very similar to the AWS values in each year. Similarly, the SW values for WRF and AWS data are very close for each year, despite larger differences in RH between WRF and AWS than between each year for each data source. Can the authors check the lines are labelled correctly and revise comments regarding differences in LW and SW.*

*Following on from the above point, section 3.2.1 needs to describe (at least in general) how short and longwave fluxes are extrapolated. From reading Liston and it seems that cloud cover is estimated using RH, then predetermined cloud extinction coefficients are used to calculate SW and LW across elevations. Was the option to 'assimilate' the observations used in this study? Neither of these is particularly standard practice when SW and LW observations are available to distribute LW and SW across a catchment, and these inputs will have a large bearing on the simulated energy and mass balances.*

**Authors' response:** MicroMet was used in 'assimilation mode' for both AWS and WRF datasets to calculate SWi and LWi , and so used available measurements to distribute these variables across the catchment. It should be noted that in the case of LWi, the data range and observed pattern compare well to observed measurements shown in the paper under review, and to measurements shown in MacDonell et al. (2013b) in the Huasco catchment (the catchment immediately to the north of the study site). It should be noted that as the area only experiences relatively low levels of cloud cover, radiative forcings are relatively "easy" to simulate. However, variables such as temperature, humidity, wind and precipitation are more complex, especially in a mountainous region. Unfortunately we only have measurements in valley floors, and so it is more difficult to assess exactly the best practice to spatially distribute these parameters.

We have included a statement in the methods section clarifying that we assimilate the data: "Radiation values for LWi and SWi are assimilated and specified using the default parameterization (Liston and Elder 2006a)."

*P13ln10 It is good to see the validation of the surface temperature (albeit at one site only), and it would be useful to include this at the beginning of section 4.2 or at least the supplementary material, given that a correct estimation of surface temperature is key to a correct simulation of melt vs sublimation. The surface temperature simulated with WRF should also be included on this figure too – I expect it will be lower than the AWS measurements given the predominance of sublimation.*

**Authors' response:** As suggested we have added the WRF results and included the surface temperature comparison in the supplementary information:

**D – Surface temperature**

[Figure]

*Figure S3: Observed and modeled surface temperature at Tapado AWS. Measurements correspond to the surface temperature computed from the outgoing long wave measured by the station. Cyan line correspond to the simulated surface temperature using the AWS-forcing and the purple one the WRF-forcing.*

*P22ln6 "This [larger turbulent fluxes in WRF vs AWS for 2014] can be explained by the larger snow cover simulated by WRF-forcing with snow cover in the entire catchment while the AWS-forcing only results in snow at higher elevations. Since the fluxes are computed over snow surfaces only, WRF increases the contribution of warmer, low elevation areas." Figure 8 shows that the differences in turbulent fluxes are similar at all elevations, which does not support this statement. In 2015, the snow cover is similar between WRF and AWS simulations, yet turbulent fluxes and sublimation are still much larger in the WRF simulations – suggesting that snow cover duration is not driving the differences in sublimation. Please remove or revise this statement.*

**Authors' response:** During the revision we realized that the text referred to Figure 8 instead of Figure 9, and have changed the text accordingly. In addition, we have adjusted the scale to facilitate the direct comparison of the four plots. In doing so, it becomes clearer that there is a difference in turbulent heat fluxes at different elevations. However, we agree that snow cover duration is not the only control, and that wind differences are likely to contribute. As we added additional explanation here, we decided it was more appropriate to include the text in the Discussion (Section 5.1.2), and so the paragraph now reads:

"In 2014 the turbulent fluxes are dominant for the WRF forcing but not for the AWS forcing (Figure 9 a,c). This can be partially explained by the larger SCA simulated by WRF-forcing with snow cover in the entire catchment while the AWS-forcing only results in snow at higher elevations. Additionally, WRF-forcing indicates colder, drier and windier condition than the AWS-forcing (Figure 4). Lower RH and higher wind speed will directly increase the latent heat flux, and potentially sublimation ratio, depending on surface temperature (see Figure S3 in Supplementary Material for surface temperature comparison)."

*P24ln26 "On the other hand biases in wind speed, incoming LW and SWi and air pressure are low for both years (results not shown)." Figure 4 now wind speed, LWi and SWi, this comment needs revised. A large bias in wind speed is shown in the figure and needs discussing.*

**Authors' response:** The comment regarding the bias in wind speed has been corrected, and section 5.1.2 has been rewritten.

*sublimation rate and ratio (especially in 2015, Figure 10)." This comment is confusing as SCD in 2015 was similar between AWS and WRF simulations, hence it cannot be the driver for differences between WRF and AWS. Please revise.*

**Authors' response:** We agree that the text is unclear, we have edited the paragraph, which now reads:

"The SCD also has a significant influence on sublimation. For 2014, the differences in SCD between the two forcings were 100 days below 4500m, and close to 50 days above this elevation (Figure 7a). Snow that persists until later in the year (austral Spring and Summer) results in an increased total melt rate and can influence the sublimation ratio (especially in 2015, Figure 10). This is the only explanation for the larger melt fraction observed with AWS-forcing compared to WRF-forcing at low elevations, given the cold bias in WRF-forcing (Figures 8c and 4b). Since a larger SCD can be related to larger precipitation amount, precipitation uncertainties likely play a significant role and the sublimation estimation."

*P25l8 "This is the only explanation for the larger melt rate observed with AWS-forcing compared to WRF-forcing at low elevations, given the cold bias in WRF-forcing (Figures 8c and 4b)." Figure 10c shows that in 2014 the low elevation areas show a similar melt rate between WRF and AWS simulations so it is unclear what is meant here. The cold bias in WRF forcing would tend to favour lower melt, which would explain the higher melt with AWS forcing. Please revise.*

**Authors' response:** The paragraph has been clarified to talk about melt fraction as opposed to rates and the impact of snow permanence until the austral Spring and Summer months. Please see previous response.

***3- Editorial comments:***

*Figure 4f,g – please check the lines are labelled correctly.*

**Authors' response:** Yes, see comment above.

*Figure 5a – should the legend read "AWS sim" rather than "WRF sim"*
*Figure 5 caption – correction to colours – "vs. observed (red)…" and "Grey shaded areas.."*

**Authors' response:** Done

*Figure 8a ylabel – "…(%) for 2014"*
*Figure 8 caption – "… forcing for 2014 (a) and 2015 (b). Simulated annual average total ablation (sublimation and melt) …*

**Authors' response:** Done

*Figure 9 – common x limits would make it easier to compare between years as well as with elevation.*

**Authors' response:** Done

*Figure 11 – common y limits would make it easier to compare between forcing datasets.*

**Authors' response:** Done

[revised manuscript text omitted]

**SUPPLEMENTARY MATERIAL**

**A – Climatology at La Laguna**

[Figure]

*Figure S1: (a) Monthly precipitation and (b) air temperature recorded at La Laguna AWS. The monthly mean*
5    *(black) is computed over the 1976-2016 period.*

**B – Comparison AWS vs. WRF grid point**

*Table S1: Statistics (R2, RMSE and AME (Absolute Mean Error)) between micromet outputs forced by the AWS*
*measurements and the WRF outputs.*

| | Vertical offset with the WRF grid point | T (°C) | | | RH | | | SWi | | | LWi | | | WS (m s⁻¹) | | | P (mm d⁻¹) | | |
|---|---|---|---|---|---|---|---|---|---|---|---|---|---|---|---|---|---|---|---|
| | | $R^2$ | RMSE | AME | $R^2$ | RMSE | AME | $R^2$ | RMSE | AME | $R^2$ | RMSE | AME | $R^2$ | RMSE | AME | $R^2$ | RMSE | AME |
| Colorado Bajo | 278m | 0.94 | 5.02 | 5.32 | 0.04 | 15.7 | 15.7 | 1 | 1.29 | 0.65 | 0.96 | 5.53 | 4.10 | 0.01 | 1.61 | 1.29 | 0.35 | 2.96 | 3.42 |
| La Laguna | 144m | 0.91 | 5.01 | 4.60 | 0.04 | 19.4 | 14.8 | 1 | 1.65 | 0.72 | 0.94 | 6.66 | 5.00 | 0.15 | 1.71 | 1.15 | 0.51 | 5.72 | 3.52 |
| Llanos de las Liebres | 305m | 0.91 | 6.39 | 6.06 | 0.04 | 20.6 | 15.8 | 1 | 1.34 | 0.51 | 0.98 | 5.57 | 4.48 | 0.37 | 2.08 | 1.46 | 0.51 | 7.17 | 4.31 |
| La Gloria | 56m | 0.89 | 5.33 | 4.81 | 0.08 | 19.7 | 15.2 | 0.99 | 2.95 | 1.09 | 0.93 | 6.61 | 5.12 | 0.17 | 1.34 | 1.01 | 0.58 | 5.05 | 3.05 |
| Colorado Alto | 196m | 0.89 | 6.61 | 6.27 | 0.04 | 21.6 | 16.6 | 1 | 1.54 | 0.53 | 0.99 | 4.46 | 3.69 | 0.52 | 2.31 | 1.70 | 0.51 | 8.35 | 4.16 |
| Vega Tapado | -145m | 0.88 | 7.01 | 6.73 | 0.03 | 22.8 | 17.6 | 0.99 | 3.43 | 1.40 | 0.99 | 4.24 | 3.67 | 0.53 | 4.28 | 3.48 | 0.53 | 8.95 | 4.73 |
| Tapado | 462m | 0.88 | 6.83 | 6.54 | 0..01 | 22.9 | 17.8 | 0.99 | 2.98 | 1.33 | 0.99 | 1.48 | 1.11 | 0.52 | 3.45 | 2.78 | 0.51 | 9.48 | 4.92 |
| Paso Aguas Negras | -121m | 0.83 | 7.29 | 6.99 | 0.00 | 24.6 | 19.0 | 1 | 1.96 | 0.75 | 0.98 | 2.96 | 2.39 | 0.54 | 5.19 | 4.27 | 0.56 | 7.89 | 4.25 |

**C- Spatial variability of the annual SCD.**

[Figure]

*Figure S2: Snow cover duration from MODIS images for (a) 2014 and (d) 2015. Simulated snow cover duration using AWS-forcing for (b) 2014 and (e) 2015. Simulated snow cover duration using WRF-forcing (c) 2014 and (f) 2015.*

**D – Surface temperature**

[Figure]

*Figure S3: Observed and modeled surface temperature at Tapado AWS. Measurements correspond to the surface temperature computed from the outgoing long wave measured by the station. Cyan line correspond to the simulated surface temperature using the AWS-forcing and the purple one the WRF-forcing.*

**E- Influence of the precipitation amount on the energy fluxes contribution**

[Figure]

*Figure S4: Monthly average (for the entire catchment, over snow surfaces only) of the main modeled heat fluxes for (a) 2014 and (b) 2015 AWS-forcings, (c) a simulation using the 2015 precipitation and the 2014 AWS-forcings, and (d) a simulation using the 2014 precipitation and 2015 AWS-forcings. SW is net shortwave radiation, LW is net longwave radiation, QE is the latent heat flux and QH is the sensible heat flux.*

*Figure S2: Monthly simulated melt (purple) and sublimation (blue) using AWS-forcing (a,b) and AWS-forcing with (c) 2014 precipitation and (d) 2015 precipitation. Forcing. Back numbers indicate the monthly sublimation ratio in %.*

**F. Model uncertainties**

The main forcing uncertainties were estimated to be the precipitation (related to P measurements and the P spatialization as there are only 2 recording stations); and the wind speed spatialization (Gascoin *et al.,* 2013). Regarding the other variables T, RH, SW, and LW, a similar study performed in a nearby catchment indicated a very good performance of MicroMet to spatialize these variables, with high correlations and low biases (for more details please refer to Gascoin *et al.,* 2013).

The principal calibration uncertainties are the topographic length scale used for the wind distribution and *z0* due to the absence of measurements to properly calibrate the model. Note that the albedo measurements have been used to calibrate the model, that is why this calibration is not considered as a main calibration uncertainty in this

study. Indeed, if we compare the model output albedo with the measurements, the mean *R* is 0.74 with mean *RMSE* of 0.22 and errors are considered related to the simple approach of the albedo computation of the model.

*Table S2: Mean annual melt and sublimation ratio of each simulation. Bold values indicate when results are different than the reference study (i.e. results present in section 4 with the AWS-forcings) indicated in the first line.*

| | 2014 | | 2015 | |
|---|---|---|---|---|
| | Melt (m.w.e.) | Sub. (m w.e.) (Sub. Ratio) | Melt (m.w.e.) | Sub. (m w.e.) (Sub. Ratio) |
| Reference sim. | 0.047 | 0.030 (42%) | 0.21 | 0.19 (48%) |
| Precipitation +10% | **0.048** | **0.029 (41%)** | **0.23** | 0.19 **(47%)** |
| Curvature 100m | 0.047 | 0.030 (42%) | 0.21 | 0.19 (48%) |
| Curvature 1000m | 0.047 | 0.030 (42%) | 0.21 | 0.19 (48%) |
| Slope 0.25 - Curv. 0.75 | 0.047 | 0.030 (42%) | 0.21 | 0.19 (48%) |
| Slope 0.75 - Curv. 0.25 | 0.047 | 0.030 (42%) | 0.21 | 0.19 (48%) |
| *Z0* = 10 mm | **0.033** | **0.049 (66%)** | **0.12** | **0.29 (72%)** |

**F1. Precipitation uncertainties**

**a) Precipitation measurements**

As mentioned in section 5.1.2 snowfall measurements in windy conditions can suffer from an undercatch bias, and corrections are generally performed using empirical relationship (e.g. MacDonald and Pomeroy, 2007; Wolff *et al.,* 2015). The strong wind gusts in this region, and especially at high elevations (Figure 3) increase measurement uncertainties. Nevertheless, in this study it was chosen not to apply any correction. First, given the lack of continuous SWE measurements, it is not straightforward to establish an empirical correction. Also, precipitation data from two rain gauges (one shielded and the second one unshielded) located in Tapado have been averaged, to reduce the random error. Note that the data from the two rain gauges is surprisingly similar (see section 2.2.1), while we would expect a larger amount of precipitation catched by the shielded Geonor sensor, in this area of strong wind speed. This suggest that either the under catch is not si important or that the shield is not that efficient.

**b) Spatialization**

The main uncertainty for the precipitation data is due to the data spatialization at catchment scale. As only two stations have been used to force the model, the uncertainty is expect to be significant due to the orographic complexity of the catchment, but cannot be evaluated based on the current available measurements.

The interpolation of precipitation in MicroMet subroutine has been done without the use of an altitudinal gradient (called precipitation adjustment factor, Liston and Elder, 2006a) as no consistent altitudinal gradients were found in the observations (results not shown). Each event is very specific (e.g. Sinclair and MacDonell, 2016) and as a result, the altitudinal lapse rate can be either positive or negative. In addition, due to low precipitation rate and the few number of events, mainly in 2014, most of the time precipitations are recorded at one station but not at the other, or there is a delay between the events recorded at both stations, which are 14 km from one another. Different altitudinal gradients (monthly *vs* event scale) were tried but the best comparisons between simulated and observed snow depths were found without considering any gradient. Considering an altitudinal lapse rate systematically leads to an over-estimation, especially at high elevations, where the snow persists until the next season.

In addition, the model simulates the wind transport only after deposition and does not consider the preferential deposition. Again, the available measurements make it difficult to estimate this uncertainty.

**c) Sensitivity test**

A sensitivity test was performed by increasing the precipitation by 10%. Results indicate that the mean annual sublimation ratio over the catchment is very similar to that of the initial simulation (Figure 2). Indeed, it decreases by 1% for 2014 and for 2015. Regarding the spatial variability (Figure 11) the difference varies between +1 and -3 %, and larger differences are observed for 2014 between 5000 and 5500 m a.s.l. and around 4600 m a.s.l. for 2015. This maximum difference can be related to changes in spatial distribution of snow when some events occur only at high elevations.

**d) Impact on model performances**

The precipitation uncertainties explain also the better performances of the model for 2015 than 2014. Indeed, the model is sensitive to patchy snow, mainly observed at low elevation. As mentioned in section 4.2.1, modeling the spatial variability of the SD is complex and results show an overall similarity of the simulated SD between some stations. When dealing with low amount of snow, patches over the catchment are more present an increase the error.

**F2. Wind uncertainties**

**a) Wind data spatialization uncertainty**

As mentioned by Gascoin *et al.,* (2013), the wind speed simulated by MicroMet model tend to be under-estimated especially at low elevation. With the exception of Paso del Agua Negra, similar results are found in this study, with largest bias found at low elevations. Indeed, the results of the cross validation test indicate an *RMSE* of at

4.1 m.s$^{-1}$ at La Laguna and a larger bias (RMSE = 7.8 m.s$^{-1}$ at Paso del Agua Negra. Differences are related to the MicroMet interpolation and can be explained by different reasons. First, The main shortcoming of the wind interpolation module is the lack of thermal winds (e.g. katabatic winds). In addition MicroMet does not take any topography into account that determines the dynamic wind direction, implying bias in the wind interpolation.

5   Nevertheless, the absence of general trends of the under-estimation (both low and high values are under-estimated and the bias strongly depend on the location and the wind speed) makes difficult to establish a relationship to correct the bias and also to evaluate the uncertainty at catchment scale.

Therefore, the number of stations used is important as increasing the number will decrease the uncertainty. The wind uncertainty has an impact on the sublimation ratio. Indeed, for instance, increasing the wind speed by 10%

10  can induces sublimation ratio changes of 40% at high elevation (i.e. where there is no melt). The wind speed is, in fact, known to directly affect the turbulent fluxes (e.g. Litt et al., 2017)

In addition, the wind speed also impacts the snow density, which directly affects the thermal conductivity of the upper snow layers (Yen, 1981). As a consequence it impacts the surface temperatures thus the turbulent fluxes.

**b) Topographic influence on wind transport**

15  In the model, the curvature length and the influence of the slope *vs.* curvature can be calibrated to consider the topographic influence of the wind transport. While the curvature length can be approximately constrained from the DEM, the relative influence of slope and curvature is more difficult to quantify a priori. As such, sensitivity tests have been performed to evaluate the impact of these parameters on the simulation.

First, the curvature length value has been varied from 100m to 1000m, and does not impact the annual mean

20  sublimation ratio at the catchment scale (Figure 2). Nevertheless the spatial variability of the differences depends on the elevation (Figure 11). Considering a larger curvature length leads to larger sublimation ratio at low elevation (in the valley) and lower one at high elevation. When choosing a larger curvature value, the simulated snow depth is larger in the valley as the snow transport is from a larger area, and increasing the snow depth decreases the sublimation ratio. Indeed, in that case the snow persists longer in spring, and warmer temperature

25  allow increasing the melt rate.

The annual mean of the sublimation ratio at the catchment scale is not sensitive to the influence of the slope *vs.* curvature either (Figure 2) when values are ranging between 0.25 and 0.75. Nevertheless, varying this influence changes the spatial distribution of snow depth. Larger snow depths (between 18 and 26%) are observed on the ridges when the influence of the slope is larger than the curvature (results not shown). As a consequence, the

30  sublimation ratio decreases by about 10 to 20% in areas with thicker snow, and can also be explained by a the persistence of the snow cover during the spring, increasing the melt rate. On the contrary the sublimation ratio is

larger on steep slopes, when the influence of the slope is set to be larger than the curvature, as this calibration allows for more snow redistribution, decreasing the mean snow depth.

The change in snow distribution is more important when changing the influence of slope vs. curvature from 0.25 to 0.75 than when changing the curvature length from 100 to 1000 m. The calibration of these parameters remains important when the sublimation is evaluated in the valley for instance but according to our results, it does not affect the sublimation ratio when evaluated at the catchment scale

**F3. Roughness value**

Increasing the roughness value by a factor 10 increases the annual mean of the sublimation ratio by 24% for 2014 and by 20% for 2015 (Table S1). Larger changes are observed around 4600 m a.s.l. for 2014 and 2015 where the difference can reach 30% (Figure 11) .

This sensitivity test highlights the strong sensitivity and the importance of choosing an accurate value to properly quantify sublimation over the catchment. Further studies are therefore recommended to calibrate this value using turbulent flux measurements such as with an Eddy Covariance System (e.g. Litt *et al.,* 2017). Nevertheless even with measurements, significant uncertainty remains due to the strong spatio-temporal variability of the snow surface roughness. While the roughness value is used as a calibrated parameter and is not an absolute physical value, it depends on the surface roughness. The roughness is expected to increase with elevation, as penitentes are commonly observed on the lower Tapado glacier (e.g. Lhermitte et al., 2014; Nicholson et al., 2016) and surrounding areas. There is therefore a strong spatial variability of the roughness value, as penitentes are not observed over the entire catchment, but mainly in the upper part. In addition, penitentes grow in size over the season (Lliboutry, 1954) leading to a strong temporal variability of the roughness.

Due to the strong sensitivity to $z_0$ and the potential for significant spatio-temporal variability of the snow surface roughness, to reduce uncertainties, a spatio-temporal evolution of $z_0$ could be envisaged. At this stage, without more measurement it is a complicated task. Further studies based on two EC measurements over the season could help to evaluate the variability.

[Figure]

*Figure S5:* *Differences in sublimation ratio per elevation range for (a) 2014 and (b) 2015, between: reference simulation and the simulation performed with a precipitation increase at the two AWS by 10% (blue); reference simulation ($z_0$=1mm) and with an increase of $z_0$ to 10mm (green); 100 and 1000m curvature length simulations (orange) and slope vs. curvature weight of 0.25 – 0.75 and 0.75 – 0.25 (black).*